# The p63 C-terminus is essential for murine oocyte integrity

Anna Maria Lena [1], Valerio Rossi[2], Susanne Osterburg[3], Artem Smirnov [1,8], Christian Osterburg [3], Marcel Tuppi[3,9], Angela Cappello[1], Ivano Amelio [1,4], Volker Dötsch [3], Massimo De Felici[2], Francesca Gioia Klinger [2], Margherita Annicchiarico-Petruzzelli[5], Herbert Valensise[6,7], Gerry Melino [1✉] & Eleonora Candi [1,5✉]

The transcription factor p63 mediates distinct cellular responses, primarily regulating epithelial and oocyte biology. In addition to the two amino terminal isoforms, TAp63 and ΔNp63, the 3'-end of p63 mRNA undergoes tissue-specific alternative splicing that leads to several isoforms, including p63α, p63β and p63γ. To investigate in vivo how the different isoforms fulfil distinct functions at the cellular and developmental levels, we developed a mouse model replacing the p63α with p63β by deletion of exon 13 in the *Trp63* gene. Here, we report that whereas in two organs physiologically expressing p63α, such as thymus and skin, no abnormalities are detected, total infertility is evident in heterozygous female mice. A sharp reduction in the number of primary oocytes during the first week after birth occurs as a consequence of the enhanced expression of the pro-apoptotic transcriptional targets *Puma* and *Noxa* by the tetrameric, constitutively active, TAp63β isoform. Hence, these mice show a condition of ovary dysfunction, resembling human primary ovary insufficiency. Our results show that the p63 C-terminus is essential in TAp63α-expressing primary oocytes to control cell death in vivo, expanding the current understanding of human primary ovarian insufficiency.

[1] Department of Experimental Medicine, University of Rome "Tor Vergata", Rome, Italy. [2] Department of Biomedicine and Prevention, University of Rome Tor Vergata, Rome, Italy. [3] Institute of Biophysical Chemistry, Center for Biomolecular Magnetic Resonance and Cluster of Excellence Macromolecular Complexes (CEF), Goethe University, Frankfurt, Germany. [4] School of Life Sciences, University of Nottingham, Nottingham, UK. [5] IDI-IRCCS, Via dei Monti di Creta, Rome, Italy. [6] Department of Surgery, University of Rome "Tor Vergata", Rome, Italy. [7] Policlinico "Casilino", Rome, Italy. [8] Present address: Ludwig Institute for Cancer Research, University of Oxford, Oxford OX3 7DQ, UK. [9] Present address: The Francis Crick Institute, London NW11ST, UK. ✉email: melino@uniroma2.it; candi@uniroma2.it

The transcription factor p63 is crucial for mouse embryo cranio-facial, skin, and limb development[1–3]. The *TP63* gene harbours two different promoters, resulting in two N-terminal isoforms (TAp63 and ΔNp63). The full-length TAp63 isoform has an N-terminal transactivation domain (TAD), while the N-terminal truncated ΔNp63 variant lacks this region but still shows transcriptional activity on certain promoters[4,5]. The ΔNp63α isoform is mainly expressed in ectodermal and endodermal-derived tissues, including the epidermis, skin appendages, some simple epithelia, and the thymus[4,6,7]. In contrast, the TAp63α isoform is present both in the male and female germlines of reproductive organs, in adult muscle tissues and in cardiac muscle precursors[8,9]. The in vivo contributions of p63 N-terminal variants to embryonic development are clearly evident upon selective ΔNp63 and TAp63 knockout (KO) mice[8,10]: striking developmental abnormalities have been observed in ΔNp63-null mice[11], in which the indispensable role of the ΔNp63 isoform in epithelial biology has been genetically demonstrated. On the other hand, TAp63 KO mice show normal development[8,12] with abnormalities in cellular senescence that prevent premature tissue ageing and defects in glucose and lipid metabolism[10]. TAp63 is constitutively expressed in primary oocytes arrested at the dyctiate stage of the meiotic prophase I within the primordial/primary follicles[8,9]. Its activation is essential to eliminate such oocytes suffering DNA-damage-induced death and in this way to maintain the integrity of the female germline; such monitoring plays a crucial role in the quality control of the female ovarian reserve[8,13,14]. Although TAp63 and yet another, N-terminally elongated isoform, GTAp63α[15,16], has also been detected in male germ cells, its role in this context has not yet been fully clarified[17–19].

Additionally, p63 can be expressed as several C-terminal isoforms, including p63α, p63β, and p63γ[3]. Whereas the functional in vivo roles of p63 N-terminal variants have been relatively well studied, the in vivo functions of p63 C-terminal variants have not yet been investigated. Indeed, by alternative splicing, at least three different C-terminal isoforms, p63α, p63β, and p63γ, are obtained in a tissue-specific manner, both for the TAp63 and ΔNp63 isoforms[4]. Only the p63α variant harbours the sterile alpha motif (SAM) domain, which is a proposed protein–protein interaction domain of still unknown function[4,20,21], and the transactivation inhibitory domain (TID), which is involved in transcriptional inhibition[22,23]. Structural and biochemical studies have demonstrated that the p63α C-terminus contains multiple regulatory elements with different functions undergoing post-translational modifications[24–26]. For instance, the TID is important for the formation of the autoinhibitory dimeric complex of TAp63α in primary oocytes. The inactive dimeric state is thereby stabilized by an interaction network between the N-terminal transactivation domain (TAD) and the TID[24,27]. After detection of DNA damage in the oocytes, TAp63α gets phosphorylated by the priming kinase Chk2 and the executioner kinase CK1, which results in disruption of the dimer and allows formation of the active tetramer[28,29]. Subsequently, active TAp63α orchestrates oocyte death by inducing apoptosis via the target genes p53-upregulated-modulator-of-apoptosis (*Puma*) and phorbol-12-myristate-13-acetate-induced protein-1 (PMAIP1, herinafter *Noxa*)[13,29,30].

Heterozygous mutations in the p63 gene are the cause of five human developmental disorders. The ectrodactyly, ectodermal dysplasia, and cleft lip/palate syndrome (EEC, Online Mendelian Inheritance in Man, OMIM 604292), is mainly characterized by ectrodactyly, ectodermal dysplasia, limb defects and cleft lip and palate (CLP), while the limb–mammary syndrome (LMS, OMIM603543) is defined by split-hand-feet malformations, cleft palate (CP), mammary-gland hypoplasia and/or nipple aplasia. Acro–dermato–ungual–lacrimal–tooth syndrome (ADULT,

OMIM103285) patients show limb malformations, finger- and toenail dysplasia, hypoplastic breasts and nipples, primary hypodontia, and loss of permanent teeth. These three syndromes can be summarized as ELA (EEC, LMS, ADULT) syndrome due to overlapping of the patient's manifestations making classification in single syndromes difficult[31–33]. In contrast to the already mentioned syndromes, isolated split-hand/foot malformation (SHFM4, OMIM605289) only affects the limb development. Ankyloblepharon-ectodermal defects-cleft lip/palate (AEC, OMIM 106260) and Rapp-Hodgkin syndrome (RHS, OMIM129400) also show overlapping clinical features and are suggested to represent the same entity (in the following AEC/RHS). Both syndromes are characterized by CLP and ectodermal dysplasia[34].

For each of the two syndromic entities, the mutations cluster in specific domains of p63, indicating a genotype-to-phenotype correlation. ELA mutations are mainly located in the DBD, suggested to inhibit DNA binding and transactivation or in the C-terminus[35], whereas AEC/RHS mutations are only present in the C-terminus, inducing aggregation and, thus, inactivation of the protein[36].

Previous work correlated *TP63* mutations to female fertility. In case of a heterozygous two nucleotides deletion (delTT1576)[37], the ovary is completely absent leading to infertility. Furthermore, there are two related, female AEC/RHS patients (delC1783)[38] described suffering from premature menopause around the age of 30 years. Recently, Tucker et al. described *TP63* defects as the cause of isolated primary ovarian insufficiency (POI). Via introduction of nonsense mutations in the C-terminal part of the SAM domain (R555*, W559*)[39,40], p63α is missing the TID. However, the underlying molecular mechanisms require further elucidation.

To investigate the global functions of the p63 variants and to understand the p63 C-terminus contribution, we generate genetically modified mice in which exon 13 was deleted (Δ13) to replace p63α with p63β. Heterozygous female mice bearing wild-type (WT) and exon 13-deleted (HET Δ13p63) alleles are sterile due to the rapid loss of the primary oocytes stockpile. Our results indicate that the p63 C-terminus plays different roles in TAp63 and ΔNp63 isoforms and that the TID inhibitory effect is required in TAp63α-expressing tissues to control cell death. In the four p63 mutations already described to affect female fertility, the TID is missing due to early stop codons in the SAM or insertion/deletion of DNA bp creating another reading frame. Here, we report the likely mechanism for these mutations leading to infertility of the female patients. Furthermore, we investigate the transactivation and oligomeric conformation of other syndromic TAp63α mutants enabling predictions of the patients' oocyte fate. Based on our experiments, we can conclude that female HET Δ13p63 mice represent an animal model for POI.

## Results

**Generation of the Δ13p63 heterozygous mice.** To investigate the roles of p63 isoforms in vivo, we decided to selectively delete *Trp63* exon 13. Exon 13 encodes the SAM domain, typical of the p63α isoform, and its genomic deletion leads to expression of the p63β isoform instead of the p63α isoform, as shown in detail by nucleotide and amino acid sequence alignments (Supplementary Fig. 1a, b). We generated an exon 13 floxed allele introducing, by homologous recombination, two loxP sites into *Trp63* introns flanking the exon 13 3' and 5' ends (Fig. 1a). Heterozygous floxed allele mice were then crossed with CMV-Cre mice to obtain the heterozygous Δ13p63 strain (HET Δ13p63, indicated as HET in figures, Fig. 1b–d). HET Δ13p63 mice were born alive at the expected Mendelian ratios (Fig. 1b) and showed normal development and morphology both at birth and in

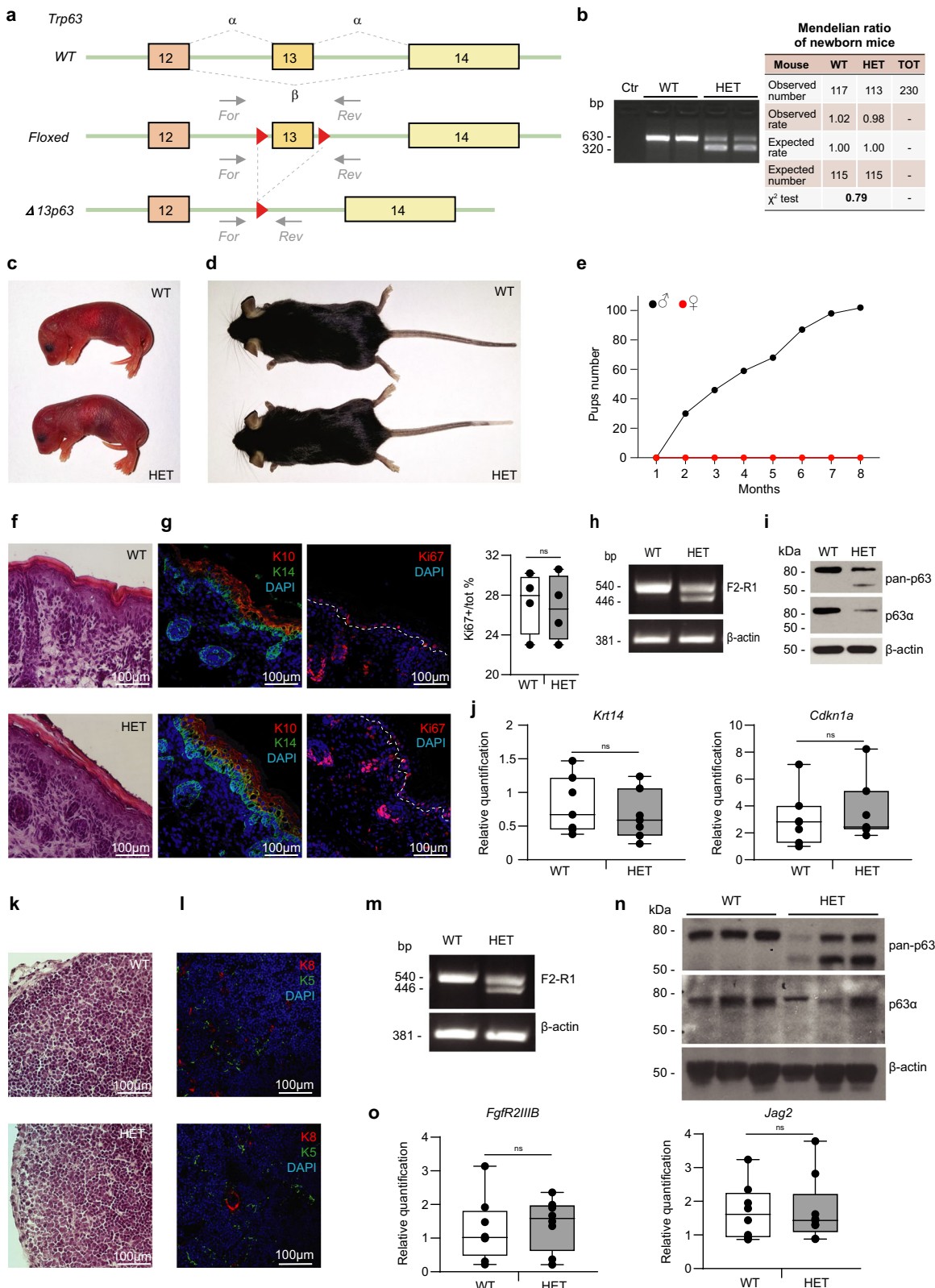

adulthood compared to wild-type (WT) mice (Fig. 1c, d). To verify that both p63α and p63β isoforms were expressed in HET Δ13p63 animals, we performed semi-quantitative PCR using cDNA derived from RNA extracted from newborn epidermis and thymus samples as a template. Exons 11, 12, and 13 were amplified using forward primers (F1, F2, and F3) and a reverse primer (R1) in the *Trp63* 3'UTR region (see Supplementary

Fig. 1a–d) to show that in p63 HET Δ13p63 animals, only exon 13 was missing in one of the two alleles. The expression of the p63γ isoform was unaltered in both WT and p63 HET Δ13p63 animals (primers F4-R2, Supplementary Fig. 1d). Sequencing of the PCR fragment after gel extraction demonstrated the correct splicing leading to p63β mRNA (Supplementary Fig. 1c). Unfortunately, HET Δ13p63 females mated with both HET

**Fig. 1 Generation and characterization of Δ13p63 heterozygous mice. a** Strategy for the generation of Δ13p63 heterozygous mice (HET Δ13p63, indicated as HET in figure). The targeting vector was generated by inserting loxP sites (red triangles) into the genomic regions flanking exon 13 to obtain a floxed allele. Heterozygous floxed mice were crossed with CMV-Cre transgenic mice to obtain HET mice. The primers used for genotyping the WT and HET mice are shown (For, forward; Rev, reverse); **b** Agarose gel electrophoresis of the PCR products obtained using genomic DNA from WT and HET mice as a template. Ctr indicates a no-template PCR. Table on the right side shows Mendelian distribution of genotypes in newborn mice (WT and HET). $p > 0.05$ by $X^2$ test; **c** Newborn WT and HET male mice; **d** Three-month-old WT and HET male mice; **e** Pups number obtained by crossing HET males (black line, $n = 4$) and HET females (red line, $n = 10$) with WT mice for a period of 8 months; **f** Hematoxylin and eosin staining of P1 backskin sections of WT and HET mice; **g** IF staining of K10/K14 and Ki67 marker on P1 backskin sections of WT and HET. Ki67 positive nuclei counting was performed on P1 backskin sections of WT ($n = 6$) and HET ($n = 6$) mice. Epidermis Ki67 positive nuclei percentage was shown in the graph on the right side. The centre of the boxplots represents median, boxes represent first (25%) and third (75%) quartiles, whiskers extend to the most extreme datapoints that are no more than 1.5-fold of the interquartile range from the box. Single values are plotted as individual points. $p$-value $> 0.05$ by two-tailed unpaired Student's $t$-test; **h** Semi-quantitative PCR analysis of p63 isoforms expression in P1 epidermises of WT and HET mice; **i** WB analysis of p63 isoforms expression in WT and HET mice protein extracts of cultured primary keratinocytes. β-actin was used as loading control. **j** Boxplots showing *Krt14* and *Cdkn1a* (p21) expression quantification by RT-qPCR in epidermises of WT ($n = 7$) and HET ($n = 7$) mice. $p$-value $> 0.05$ by two-tailed unpaired Student's $t$-test; **k** Haematoxylin and eosin staining of P1 thymus sections of WT and HET mice; **l** IF staining of K5/K8 markers on thymus sections of WT and HET mice; **m** Semi-quantitative PCR analysis of p63 isoforms expression in P1 thymus of WT and HET mice; **n** WB analysis of p63 isoforms expression in protein extracts of WT and HET mice P1 thymuses. β-actin was used as loading control. **o** Boxplots showing *FgfR2IIIb* and *Jag2* expression quantification by RT-qPCR in thymuses of WT ($n = 7$) and HET ($n = 7$) mice. $p$-value $> 0.05$ by two-tailed unpaired Student's $t$-test. The images shown are representative of all the experiments performed (at least $n = 6$) Source data are provided as a Source Data file.

Δ13p63 and WT males never became pregnant over a period of 8 months, excluding the possibility of obtaining Δ13p63 homozygous animals (Fig. 1e). HET Δ13p63 males were fertile, and when crossed with WT females, they generated litters with normal numbers and Mendelian genotype ratios (Fig. 1c), indicating that the HET Δ13p63 female mice have selectively impaired fertility.

**Δ13p63 heterozygous mice develop normal skin and normal thymus**. To understand the role of p63α in ΔNp63-expressing tissues, we evaluated the skin and thymus from newborn mice, two organs in which ΔNp63α is expressed at high levels and plays a functionally relevant role[6,7,41,42]. Histological characterization by haematoxylin-eosin (H/E) staining showed normal structures of the dorsal epidermis (Fig. 1f). The expression of the basal layer marker keratin (K) 14 (K14) and that of the differentiation layer marker K10 were identical between WT and HET Δ13p63 mice, as was the expression of the proliferation marker Ki67 (Fig. 1g), suggesting normal proliferation and differentiation programmes in the HET Δ13p63 epidermis. In the epidermis, the isoforms (ΔNp63α and ΔNp63β) were expressed at 1:1 ratio at the mRNA level (Fig. 1h, Supplementary Fig. 1d). Western blot analysis also showed their expression, as indicated by staining of epidermal extracts with an anti-pan-p63 antibody (Fig. 1i). Furthermore, no differences were detected by Real Time (RT)-qPCR in the expression of two bona fide ΔNp63α skin-specific target genes, *Krt14* and *Cdkn1a* (p21)[6,43–45] (Fig. 1j). These results further confirmed that the simultaneous expression of ΔNp63α and ΔNp63β in the HET Δ13p63 epidermis does not affect normal function during development, possibly because the two variants act synergistically to transcribe the same target genes.

Similar results were obtained in the thymus, in which ΔNp63α, expressed in thymic epithelial cells (TECs), sustains thymus development through the regulation of two target genes, *FgfR2IIIb* and *Jag2*[41,46–48]. Histological characterization by haematoxylin-eosin staining showed normal organ structure (Fig. 1k). No differences were detected in the expression of the TEC differentiation markers K8 and K5 (Fig. 1l). Both isoforms (ΔNp63α and ΔNp63β) were detected in the thymus at the mRNA level (Fig. 1m, Supplementary Fig. 1d) and at the protein level (Fig. 1n), as indicated in the Western blot obtained using the anti-pan-p63 antibody. Additionally, no differences were detected by RT-qPCR in the expression of two bona fide thymus-specific ΔNp63α target genes, *FgfR2IIIb* and *Jag2*[41] (Fig. 1o). Taken

together, these data indicated that in ΔNp63-expressing organs, the thymus and epidermis, the simultaneous expression of ΔNp63α and ΔNp63β did not alter the organ development or the expression of p63 tissue-specific target genes.

**Δ13p63 heterozygous females show primary ovary insufficiency**. TAp63 isoforms are expressed both in male and female germlines[8,15,16]. During mouse ovary development, TAp63α starts to be expressed in primary oocytes around embryonic day 17 during the prophase of the first meiotic division[8]. At 4–7 days post-partum (dpp), all oocytes arrested at the dyctiate stage of the meiosis within primordial/primary follicles are TAp63α positive[8]. TAp63 is dispensable for ovarian development, as shown in TAp63 KO mice[12]; nevertheless, it plays a crucial role in the quality control of the primary oocytes being activated and inducing apoptosis in those presenting DNA damage[8,13,14]. Therefore, to understand the infertility phenotype observed in the heterozygous females, a deeper analysis of the ovarian tissues was carried out. Macroscopic analysis of postnatal day 45 (P45) ovaries from WT and HET Δ13p63 females revealed evident organ size difference: postnatal day 45 ovaries from HET Δ13p63 mice were significantly smaller than those from WT mice (Fig. 2a). Through semi-quantitative RT-qPCR analysis of TAp63 expression in ovaries from WT and HET Δ13p63 females from day 17.5 of embryonic development (E17.5) to postnatal days 1 (P1), 3 (P3), 7 (P7), and 10 (P10), we confirmed that in WT mice TAp63α isoform is detectable at E17.5 and that its expression remained high in primordial and primary follicles (P1–P10; Fig. 2b). On the other hand, in HET Δ13p63 ovaries, while there was the expected increase from E17.5 to P1 (Fig. 2b), at later time points, both p63α and p63β mRNA levels decreased along with a reduction in oocytes number (Fig. 2d–g), which appeared associated to increased apoptosis (Fig. 3). Since within the ovaries, p63 is exclusively expressed in the oocytes, depauperation of these population resulted in reduction of p63 mRNA. In HET Δ13p63 ovaries, p63γ isoform and p53 mRNAs followed similar trend (Supplementary Fig. 1e). Western blot analysis showed that only the TAp63α isoform was detectable at very low levels in P1 HET Δ13p63 ovary extracts, while the levels of the TAp63β variant were not appreciable (Fig. 2c). Mechanistically, however, we also proved that active TAp63β variant undergoes a high proteosomal degradation rate (Supplementary Fig. 2a–c). Thus, both oocytes depauperation and poor protein stability underlyed low expression level of TAp63β in HET Δ13p63 ovary extracts. Histological

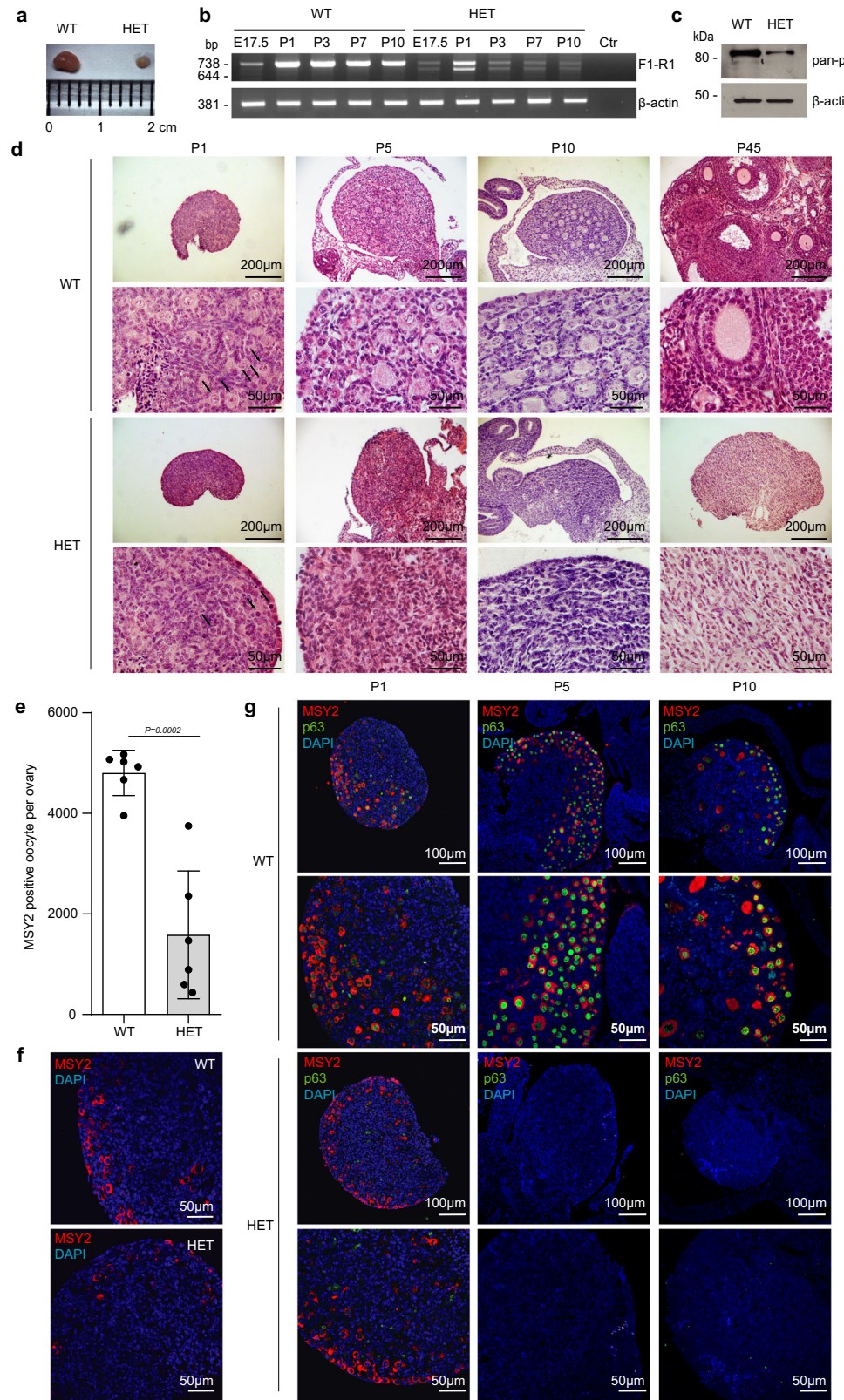

characterization by H/E staining of P1, P5, P10, and P45 ovaries revealed that, while WT and HET Δ13p63 P1 ovaries showed similar morphologies with the presence of primordial follicles (Fig. 2d, arrows), HET Δ13p63 P5 and P10 ovaries presented an atretic morphology and almost total absence of follicles, aspects even more evident in HET Δ13p63 P45 ovaries (Fig. 2d). More-over, HET Δ13p63 P1 ovaries showed significant less (40%)

numbers of oocytes identified by MSY2 compared to WT P1 ovaries (Fig. 2e, f). Double IF co-staining of MSY2 and p63 showed that in WT ovaries oocytes were positive for both as expected, while in HET Δ13p63 ovaries, oocytes were mainly stained for MSY2 (Fig. 2g). These observations strongly support the hypothesis that oocytes depletion in heterozygous ovaries is correlated with the activity of TAp63β isoform, that is known to

**Fig. 2 Characterization of Δ13p63 heterozygous mouse ovaries. a** P45 WT and HET Δ13p63 (indicated as HET in figure) ovary size; **b** Semi-quantitative PCR analysis of p63α and p63β isoforms expression E17.5, P1, P3, P7, and P10 ovaries of WT and HET mice. β-actin was used as housekeeping gene for normalization. Ctr indicates a no-template PCR; **c** WB analysis of p63 isoforms expression in protein extracts of P1 ovaries of WT ($n = 3$) and HET ($n = 3$) mice. β-actin was used as a loading control; **d** Haematoxylin and eosin staining at P1, P5, P10, and P45 (1, 5, 10, and 45 day post-partum, P) ovary sections of WT and HET mice. The panels on the right side are magnifications of areas of the panels on the left side. Arrows indicate oocytes; **e** MSY2 positive cell count/ovary; WT ($n = 6$) and HET ($n = 6$) ovaries. Data are presented as mean ± SD, *p*-value by two-tailed unpaired Student's *t*-test; **f** Oocytes IF staining for MSY2 (red) in P1 ovary sections of WT and HET mice; **g** MSY2 and p63 co-staining of P1, P5, and P10 ovary sections of WT and HET mice. The panels on the right are magnifications of areas of the panels on the left side. The images shown are representative of all the experiments performed (at least $n = 6$) Source data are provided as a Source Data file.

be rapidly degraded (Supplementary Fig. 2a–c)[23]. Since TAp63 has also been detected in testis[18], we compared P45 WT and HET Δ13p63 testis. No abnormalities were observed at the gross morphology level. Morphology, as evaluated by PAS staining, did not reveal differences between WT and HET Δ13p63 testis (Supplementary Fig. 3a, b). Both TAp63α and TAp63β mRNA variants were expressed at comparable levels (Supplementary Fig. 3c). Taken together, these results demonstrate that the simultaneous expression of TAp63α and TAp63β causes a condition of ovary disfunction resembling primary ovary insufficiency (POI) in human females but does not affect testis morphology or function.

**Δ13p63 heterozygous primary oocytes undergo uncontrolled apoptotic death.** To investigate in vivo the underlying molecular events driving HET Δ13p63 infertility, we crossed HET Δ13p63 males with transgenic p70-c-Kit/GFP females[49] to obtain p70-c-Kit/GFP/Δ13p63 heterozygous mice (also named WT-GFP or HET Δ13p63-GFP mice). These mice showed green fluorescence only in primary oocytes, enabling us to follow their primary oocyte postnatal development ex vivo from P1 to P7. WT and HET Δ13p63-GFP ovaries were isolated and cultured ex vivo, as shown in Fig. 3a. Confocal scanning of whole ovaries with z-stacks acquisition every 5 μm confirmed that under ex vivo conditions, apoptotic death occurs in a constitutive manner during primary follicle development without inducing damage (Fig. 3a). HET Δ13p63-GFP ovaries, compared with WT ovaries, progressively and rapidly lost green fluorescent oocytes from P1 to P7; at day 7, the oocytes became undetectable (Fig. 3a). These striking results suggest that the oocytes disappeared due to an accelerated uncontrolled cell death programme mediated by TAp63β. To test whether the primary oocytes died by overt apoptosis, we added the pan-caspase inhibitor Z-VAD to the HET Δ13p63-GFP ovary culture medium. Since, distinguishing individual oocytes immediately after birth (P0-P3) is quite challenging, making virtually unreliable counts, we quantified the intensity of GFP fluorescence to estimate the number of oocytes in different conditions. Indeed, intensity of GFP is proportional to the number of oocytes, as assessed by manual counting of oocytes at P4 where they are clearly separated individually. After Z-VAD treatment, we observed a significant delay in cell death, as indicated by the rescue of the green fluorescent oocytes in Z-VAD-treated HET Δ13p63-GFP ovaries compared to control DMSO-treated HET Δ13p63-GFP ovaries (Fig. 3b, c). Accordingly, we observed significant upregulation of apoptotic TAp63 target genes involved in primordial follicle cell death, namely, *Puma* and *Noxa*, in HET Δ13p63 compared to WT ovary samples (Fig. 3d). These results suggest that the expression of TAp63β is sufficient to trigger uncontrolled apoptosis in primary oocytes independent of exogenous damage.

**TAp63β forms constitutively active tetramers.** TAp63α's presence from E17.5 to P10 in mouse ovaries does not impair normal

follicle development in contrast to TAp63β. In its inactive state, TAp63α adopts a closed, dimeric conformation, mediated by the transactivation (TA) and transcription inhibitory (TID) domains (Fig. 4a). In case of DNA damage, TAp63α plays a protective quality control role, triggering cell death in damaged oocytes[8]. Specific residues in the linker region between the SAM and TID get phosphorylated, resulting in conformational changes leading to a tetrameric active state[29]. As an active tetramer, TAp63α transcribes *Puma* and *Noxa* target genes to induce cell death[8,14]. This regulatory mechanism is unique to TAp63α and essential for formation of inactive dimers. Consequently, in contrast to WT TAp63α, deletions in TAp63α C-terminus [(…572), (…511), (…410)] missing the TID, TAp63β, and TAp63γ showed high transactivation in an in vitro transcription luciferase assay using the PUMA-responsive element in agreement with earlier reports (Fig. 4b)[4,22]. To note, the variants with higher transcriptional activity are the one detected less efficiently by western blot (Fig. 4b). As expected and shown by blue native PAGE (BN-PAGE) analysis, TAp63β and TAp63γ, constitute tetramers, while TAp63α is kept as a dimer (Fig. 4c, Supplementary Fig. 4a). These findings confirm that the activity of TAp63 is exclusively regulated by the oligomeric state, switching from inactive dimer to constitutively active tetramer via phosphorylation, during DNA damage in oocytes[29], or lack of the TID, which is the case for all C-terminal isoforms except p63α. These data also are in agreement with the observation that only TAp63α but not TAp63β can be detected at the protein level (Fig. 2c, g) since transcriptionally active p63 isoforms are quickly degraded while inactive ones accumulate[22,50].

In HET Δ13p63 mouse p63α and p63β are expressed simultaneously, in skin as ΔNp63 and in oocytes as TAp63, respectively. To investigate a potential interplay between both C-terminal isoforms, we performed in vitro transcription luciferase assays using PUMA- and NOXA-responsive elements[51,52]. Co-expression of TAp63α and TAp63β did not reduce the measured overall activity compared to only TAp63β, suggesting that TAp63α cannot interfere with TAp63β transcriptional activity (Fig. 4d). This is consistent with the findings that the closed dimer of TAp63α does not interact or oligomerize with open tetramers[27]. In contrast, ΔNp63α and ΔNp63β, expressed as tetramer (Fig. 4c), were both active when tested using envoplakin (ENV)- and bullous pemphigoid antigen 1 (BPAG1)-responsive elements[53] (Fig. 4e). Control Western blots of luciferase assay extracts are shown in Supplementary Fig. 5a, b.

Taken together, these results support the finding that while ΔNp63β synergistically acts with ΔNp63α during epithelial development, TAp63β isoform is able to induce uncontrolled oocyte cell death independently of DNA-damage-driven phosphorylation and activation signals, causing POI in HET Δ13p63 female mice.

**Mutant TAp63α's oligomeric state determines oocyte fate.** The results with the HET Δ13p63 mouse described above have also immediate implications for the fertility of human patients suffering from different p63-related syndromes. Currently, four

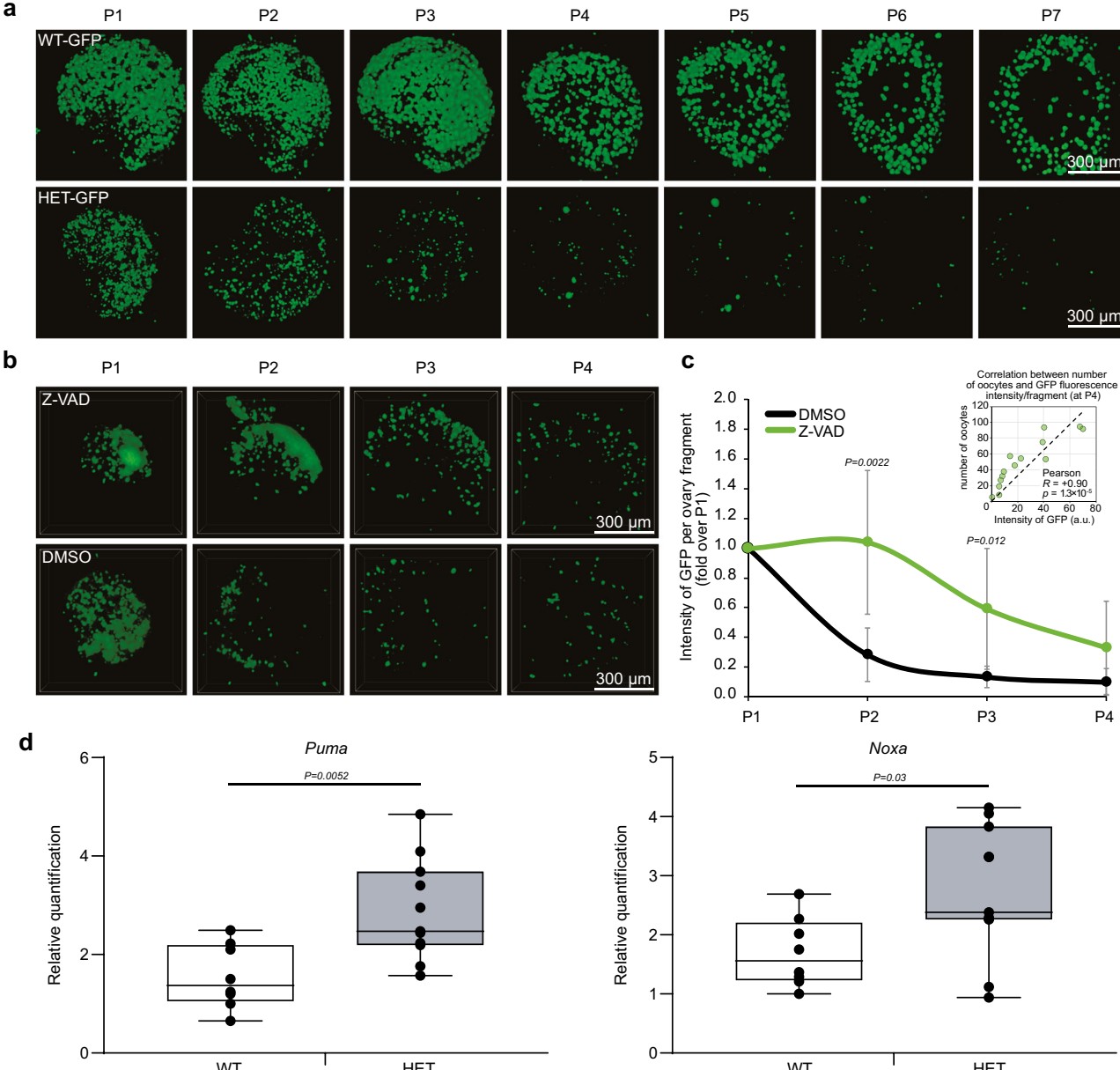

**Fig. 3 The Δ13p63 heterozygous oocytes died by apoptosis. a** Time-course (P1–P7) confocal imaging of the entire ovaries from WT-GFP and HET Δ13p63-GFP (indicated as HET-GFP in figure) mice showing 3D rendered reconstructions using an alpha-blending algorithm. Oocytes were labelled by the expression of p-70/c-Kit fused to GFP. Images were acquired with a step distance of 5 μm. **b** Confocal imaging of ovarian fragments from heterozygous mice, performed as described in **a**. Ovaries were divided into two fragments at P1 and treated with either DMSO or 50 μM Z-VAD for three days. **c** Intensity of GFP/ovary fragment at P1, P2, P3, P4 in presence or absence of Z-VAD (50 μM). n = 7 (DMSO) and n = 7 (Z-VAD) ovary fragments were analysed. Data are presented as mean ± SD, *p-value by two-tailed unpaired Student's t-test. Correlation between GFP fluorescence intensity and oocytes number (P4) is shown in the scatterplot in the upper right corner. Pearson correlation (R), between GFP fluorescence intensity/fragment and oocytes number (P4) and p-value are shown (n = 7 (DMSO) and n = 7 (Z-VAD) ovary fragments); **d** Boxplots showing Puma and Noxa expression quantification by RT-qPCR in P1 ovaries of WT (n = 9) and HET (n = 9) mice. The centre of the boxplots represents median, boxes represent first (25%) and third (75%) quartiles, whiskers extend to the most extreme datapoints that are no more than 1.5-fold of the interquartile range from the box. Single values are plotted as individual points. p-value by two-tailed unpaired Student's t-test. The images shown are representative of all the experiments performed (at least n = 3) Source data are provided as a Source Data file.

different mutations have been reported to affect female fertility causing either POI (R555*, W559*)[39,40], premature menopause (delC1783)[38] or complete absence of ovaries (delTT1576)[37]. The common feature of those frameshifts and nonsense mutations is the heterozygous loss of the TID, similar to our HET Δ13p63 mouse model. Within the ELA and AEC/RHS syndrome families many more mutations affecting the p63 C-terminus have been identified, but so far their effect on female fertility has not been

investigated (Fig. 5a). To correlate the effect of the p63 mutations with the likelihood for infertility of female patients we have measured the transcriptional activity in vitro of several mutants belonging to ELA, the AEC/RHS or SHFM syndrome in reporter luciferase assays (Fig. 5b, Supplementary Fig. 4b), also including R555*, W559*, delC1783, and delTT1576.

ELA syndrome mutations can be subdivided into two different categories: point mutations within the DNA-binding domain

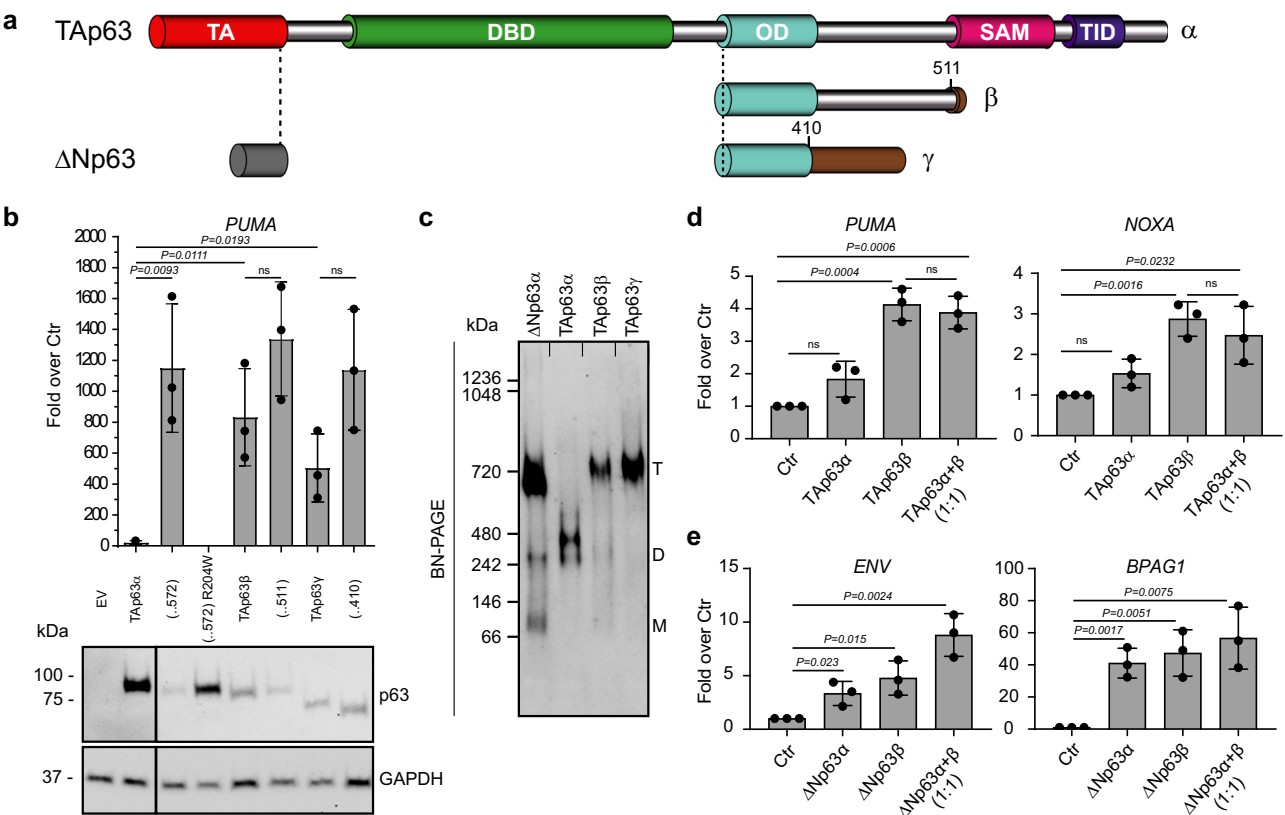

**Fig. 4 The TAp63β isoform is a transcriptionally active tetramer. a** TAp63 and ΔNp63 isoforms domains: TA transactivation domain, DBD DNA-binding domain, OD oligomerization domain, SAM sterile alpha motif, TID transactivation inhibitory domain. The C-terminus of p63β and p63γ diverge from p63α starting from residue 411 or 512, respectively. **b** The relative luciferase activity expressed as the fold over control (Ctr) of TAp63 isoforms shows that specific C-terminus amino acids do not reduce transcriptional activity except the TID in case of the α-isoform. TAp63 (..572) misses the post-SAM part of the protein; TAp63 (...411) reflect the common part of TAp63β; and TAp63γ with TAp63α. n = 3 biologically independent experiments, luciferase assay was performed in triplicates on *PUMA* 4xBS2WT responsive elements (REs). Data are presented as mean ± SD, *p*-value by two-tailed unpaired Student's *t*-test. WB analysis of TAp63 isoforms and deletion variants in luciferase assay extracts are shown below the graph. GAPDH was used as loading control; **c** Oligomeric state analysis of the indicated p63 isoforms via BN-PAGE expressed in vitro in rabbit reticulocyte lysate. The oligomeric conformation is indicated by T (tetramer), D (dimer), M (monomer); **d** The relative luciferase activity expressed as fold over control (Ctr) shows the ability of TAp63α and β to transactivate the *PUMA* and *NOXA* promoters. n = 3 biologically independent experiments, luciferase assay was performed in triplicates. Data are presented as mean ± SD, *p*-value by two-tailed unpaired Student's *t*-test; **e** The relative luciferase activity expressed as the fold over control (Ctr) shows the ability of ΔNp63α and ΔNp63β to transactivate the Envoplakin (ENV) and Bullous pemphigoid antigen 1 (BPAG1) promoters. n = 3 biologically independent experiments, luciferase assay was performed in triplicates. Data are presented as mean ± SD, *p*-value by two-tailed unpaired Student's *t*-test. Source data are provided as a Source Data file.

(DBD) that impair interaction with DNA (e.g. R304W) and deletions or insertions of DNA bp in the C-terminus of the protein leading to frameshifts (FSs) deleting parts of the C-terminus including the TID. As expected, R304W shows no induction in the reporter luciferase assay since this mutant is still dimeric (Fig. 5b). However, also in the tetrameric state it is not able to bind to DNA as already shown for tetrameric ΔNp63α[35]. In contrast, the ELA-FS mutations (insA1572, delTT1576, delAA1743, and delACTT1217) are transcriptionally active like TAp63β, consistent with the absence of the inhibitory TID. Investigation of the oligomeric state further confirmed that all ELA-FS mutants are tetramers (Fig. 5c).

For the AEC mutations, the situation is more diverse. This syndrome is caused either by point mutations within the SAM (e.g. L514F) or TID or by FS mutations resulting in different C-termini[20]. The common disease mechanism of all these mutations is that they expose or create aggregation prone peptide sequences that lead to aggregation and functional impairment of p63[36]. The transcriptional activities of L514F within the SAM domain and FS mutations C-terminal to the TID (delA1859) are low and

comparable to wild-type TAp63α (Fig. 5b), consistent with the formation of a closed dimeric state as observed by native BN-PAGE (Fig. 5d). The remaining AEC/RHS-FS mutations (ΔExon11, insA1456, delG1697, delA1709, delC1721, delC1729, delC1742, delC1783, delG1787), however, are tetrameric, yet show a low transcriptional activity (Fig. 5b, d, e). All of the AEC/RHS-FS mutations create new AA sequences with a high predicted aggregation propensity analysed by TANGO algorithm (Supplementary Fig. 6a–c). Nine of the eleven investigated AEC/RHS-FS result in the same very C-terminal sequence (Supplementary Fig. 6c). The two AEC/RHS point mutations in the TID (R598L, D601V) show a drastic increase in aggregation propensity compared to the wild-type (Supplementary Fig. 6a).

POI mutations R555* and W559* behave similar to the tetrameric AEC/RHS-FS in the assays. They show a relatively small induction in the reporter luciferase assay compared to TAp63β and the ELA-FSs but are tetrameric (Supplementary Fig. 6b, c). Due to the early stop codon in the SAM-domain sequence, this domain might not fold properly leading to exposition of aggregation prone regions (APRs) normally covered

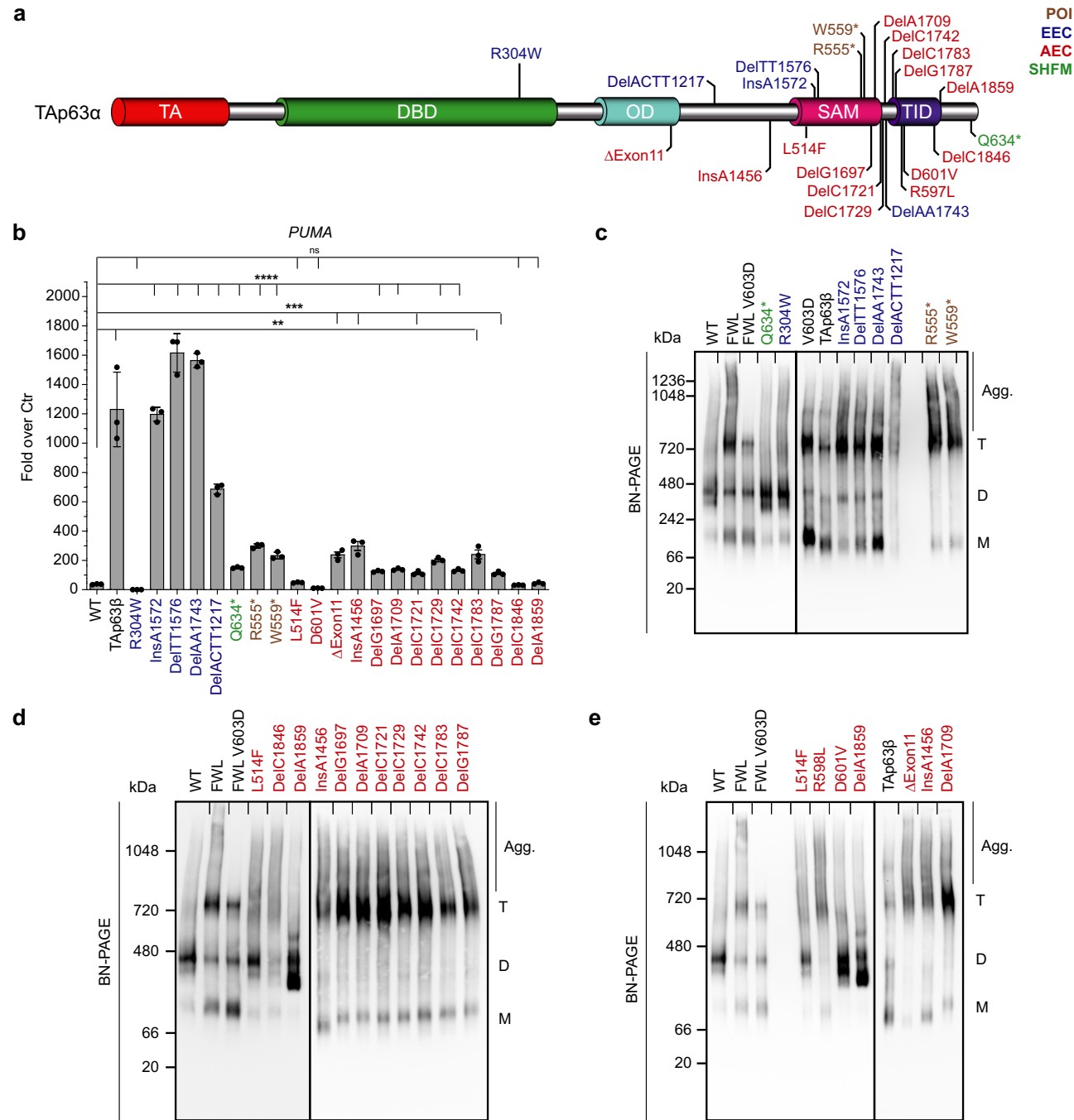

**Fig. 5 Analysis of TAp63α mutants' transactivation activity and oligomeric states. a** Overview of different syndromic mutations on TAp63α. ELA mutations are represented in blue, AEC/RHS in red, SHFM in green and POI mutations in brown. **b** The relative luciferase activity expressed as fold over control (Ctr) shows the ability of wild-type or mutant p63 variants to transactivate the *PUMA* promoters. $n = 3$ biologically independent experiments, luciferase assay was performed in triplicates on *PUMA* 4xBS2WT responsive elements (REs). Data are presented as mean ± SD, **p-value < 0.01, ***p-value < 0.001, ****p-value < 0.0001, ns p-value > 0.05 by two-tailed unpaired Student's t-test. Exact p-values are provided in Source Data file; **c–e** Oligomeric state analysis of mutant TAp63α via blue native (BN)-PAGE. The oligomeric conformation is indicated by T (tetramer), D (dimer), M (monomer) and, additionally, Agg. for aggregation. Source data are provided as a Source Data file.

in the hydrophobic core of the SAM. To confirm this suggestion, we mutated the aggregation prone regions by either introducing aspartic acid residues (mutAPR) or by simply deleting the region (ΔAPR) in the two p63α POI and selected AEC/RHS mutants, analyzed by TANGO algorithm (Supplementary Fig. 6d–g). The results of the transactivation assays show that all tetrameric AEC/RHS or POI mutants reach high transcriptional activity when aggregation is suppressed (Supplementary Fig. 7a, b) like TAp63β

and the ELA-FS mutations. On BN-PAGE, strongly reduced aggregation is visible in case of the rescue deletions (Supplementary Fig. 7c, d).

These results show that FS mutations disrupting the inactive dimeric state of TAp63α are likely to cause POI condition in female patients suffering from developmental defects in epithelial tissues and appendages caused by the same mutations in the ΔNp63α isoform. However, there is a significant difference

**Table 1 Summary table of predicted oocyte fate.**

| Mutations | Oligomeric conformation | DNA binding | Transcriptional activity | Aggregation | Predicted oocyte fate |
|---|---|---|---|---|---|
| Wild-type | | | | | |
| TAp63α | Dimer | (−) | (−) | (−) | Protected |
| TAp63α | Tetramer | (+++) | (+++) | (−) | Protected |
| TAp63β | Tetramer | (+++) | (+++) | (−) | Apoptosis |
| ELA mutants | | | | | |
| R304W | Dimer | (−) | (−) | (−) | Unprotected |
| R304W | Tetramer | (−) | (−) | (−) | Unprotected |
| DelACTT1217 | Tetramer | (+++) | (+++) | (−) | Apoptosis |
| InsA1572 | Tetramer | (+++) | (+++) | (−) | Apoptosis |
| DelTT1576 | Tetramer | (+++) | (+++) | (−) | Apoptosis |
| DelAA1743 | Tetramer | (+++) | (+++) | (−) | Apoptosis |
| SHFM mutant | | | | | |
| Q634* | Dimer | (−) | (−) | (−) | Protected |
| Q634* | Tetramer | (+++) | (+++) | (−) | Protected |
| POI mutants | | | | | |
| W559* | Tetramer | (+) | (+) | (+) | Apoptosis |
| R555* | Tetramer | (+) | (+) | (+) | Apoptosis |
| AEC/RHS mutants | | | | | |
| L514F | Dimer | (−) | (−) | (+) | Unprotected |
| L514F | Tetramer | (+) | (+) | (+) | Unprotected |
| D601V | Dimer | (−) | (−) | (+) | Unprotected |
| D601V | Tetramer | (+) | (+) | (+) | Unprotected |
| DelA1859 | Dimer | (−) | (−) | (+) | Unprotected |
| DelA1859 | Tetramer | (+) | (+) | (+) | Unprotected |
| ΔExon11 | Tetramer | (+) | (+) | (+) | Apoptosis |
| InsA1456 | Tetramer | (+) | (+) | (+) | Apoptosis |
| DelG1697 | Tetramer | (+) | (+) | (+) | Apoptosis |
| DelA1709 | Tetramer | (+) | (+) | (+) | Apoptosis |
| DelC1721 | Tetramer | (+) | (+) | (+) | Apoptosis |
| DelC1729 | Tetramer | (+) | (+) | (+) | Apoptosis |
| DelC1742 | Tetramer | (+) | (+) | (+) | Apoptosis |
| DelC1783 | Tetramer | (+) | (+) | (+) | Apoptosis |
| DelG1787 | Tetramer | (+) | (+) | (+) | Apoptosis |
| DelC1846 | Tetramer | (+) | (+) | (+) | Apoptosis |
| R598L | Tetramer | (+) | (+) | (+) | Apoptosis |

Overview of predicted oocyte fate for syndromic TAp63α mutants and TAp63β based on their oligomeric state, transactivation potential and aggregation propensity. (+++) indicates strong, (+) low, and (−) no activity, aggregation or predicted DNA binding.

between tetrameric ELA- and AEC/RHS-p63α mutations. The low activity of AEC/RHS mutations due to aggregation reflects the temporally delayed effect of premature menopause while the tetrameric ELA-FS, like TAp63β in the HET Δ13p63 mouse, would probably kill all oocytes shortly after their expression. The effect of mutations not disturbing the inactive dimer (R304W, L514F, delA1859, and Q634*) will most likely only emerges once TAp63α gets activated by DNA damage. Inhibition of DNA binding or aggregation could then delay or impair apoptosis, cancelling TAp63α's role as the guardian of the oocyte genome. An overview of these results and the predictions for the individual syndromes is summarized in Table 1.

## Discussion

The roles of TAp63 and ΔNp63 in vivo have been previously investigated by generating selective knockout mice for either isoform, resulting in the identification of distinct pheno-types[11,12,54]. However, aside from early controversial work on exon 12[25], the specific role of the p63 C-terminus in vivo remains largely unexplored. The p63α isoform is by far the most abundant p63 isoform in epithelial tissues (ΔNp63) and in primary oocytes (TAp63). Here, using a genetic mouse model, we knocked out the p63α isoform, replacing it with p63β. The results showed a pro-found phenotype in heterozygous females leading to a condition of ovary disfunction resembling POI in human females,

consistent with descriptions of p63 mutations in human patients affecting fertility (Fig. 6)[37,38,40].

Recently, a whole-genome sequencing investigation of 13 POI patients demonstrated that TAp63 truncated variants in the terminal exon 14 are associated with female infertility[40]. Both variants identified in this study, located in the SAM (R555* and W559*), generate a p63-truncated isoform lacking the TID and the C-terminal part of the SAM. This truncation is responsible for the inability of the SAM domain to adopt its native structure exposing two aggregation prone regions usually hidden in the fold. The POI mutants are tetrameric but show a limited activity due to APR exposure compared to TAp63β. The reduced activity results in a slow depletion of the oocyte pool over time. The absence of a typical AEC syndrome phenotype of the patients, however, cannot be explained.

In contrast to the POI mutants, the AEC/RHS-FS mutations create new aggregation peptides not present in the wild-type protein. The FS mutation delC1783, described to induce pre-mature menopause around the age of 30 years[38], shares the same C-terminal frame with eight of the eleven investigated AEC/RHS-FS mutants. For all tetrameric FS mutants, we would expect a similar oocyte fate of slow oocyte elimination by apoptosis. Further effects on female fertility are not described for AEC/RHS patients. On the contrary, the tetrameric ELA-FS mutants are likely to result in complete destruction of all oocytes before puberty. The results of the reporter luciferase assay suggest that

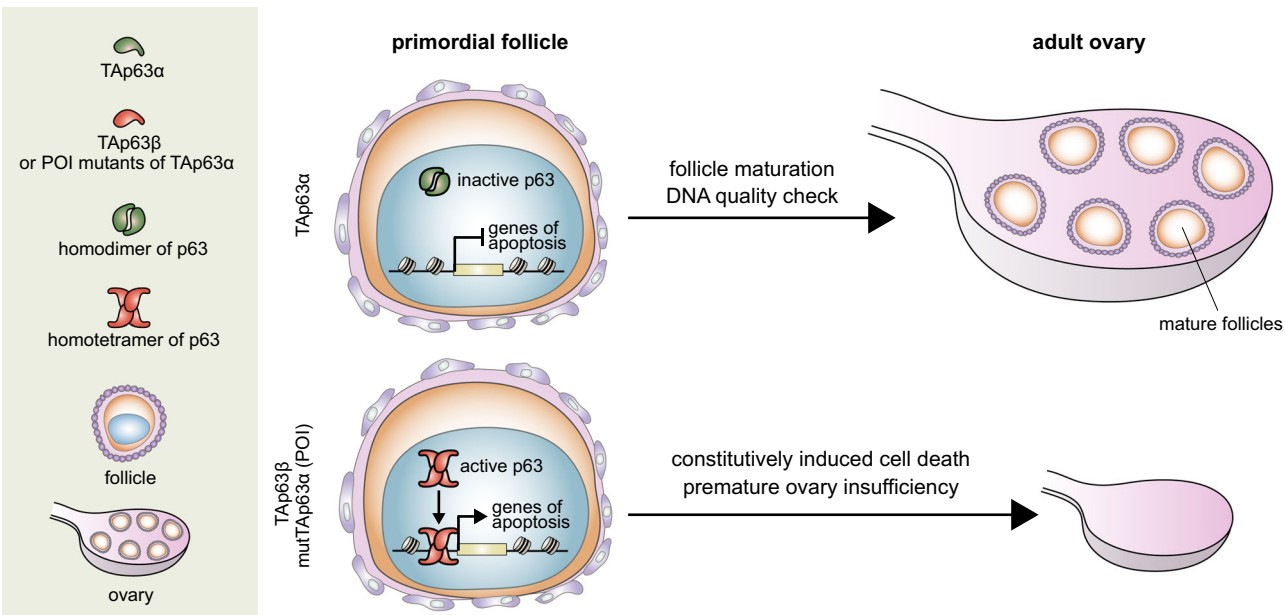

**Fig. 6 Schematic model of TAp63 role in ovary physiology, in HET Δ13p63 mouse model and in presence of POI mutations.** Under normal physiological conditions, TAp63α is present in dimeric inactive form, which can be activated only upon certain circumstances such us upon DNA damage. Activated TAp63α enables DNA quality check to ensure the genomic stability of germline ovary development and follicle maturation. However, TAp63β, as well as POI mutants of p63, form active tetramers which constitutively induce cell death, leading to premature ovarian insufficiency.

they induce uncontrolled expression of proapoptotic genes, initiating apoptosis of the oocyte similar to TAp63β.

In case of syndromic, still dimeric TAp63α mutants (R304W, delA1859, D601V, and Q634*), the survival of the oocytes should not be influenced. However, in presence of DNA damage, the quality control mechanism of TAp63α can be delayed or completely disturbed. After phosphorylation, TAp63α mutants cannot initiate directly transcription of proapoptotic genes because they are either incapable to bind to DNA or their activity is suppressed by protein aggregation.

Several studies have reported the detection of p63 (and in particular TAp63α) in male germ cells. Mice with a Trip13 mutation, a conserved AAA + ATPase required for the completion of meiotic DSB repair, arrest spermatocytes at the pachytene stage of meiosis caused by unrepaired DNA double-strand breaks. Inactivation of either p53 or TAp63 in these mice, however, enables spermatocytes to further progress while inactivation of the third family member, p73, has no effect. It was further shown that this pachytene cell-cycle arrest in spermatocytes of Trip13 mutant mice can be prevented by eliminating the kinase ATM or the ATM-effector kinase Chk2 which in the female germline are essential for the activation of TAp63α[55,56]. Despite these previous findings, our HET Δ13p63 mouse does not show an effect on male fertility. If TAp63α in male germ cells constitutes a similar quality control check point as in female germ cells the expression of the constitutively active TAp63β should eliminate all spermatocytes passing through the pachytene check point. The uncompromised fertility of the male HET Δ13p63 mice raises questions about the role of TAp63α during spermatogenesis and calls for a more detailed analysis including the quantity and quality of sperm cells produced in these mice.

In humans and great apes yet another isoform of p63 has been identified, called GTAp63α. This isoform is created by the by insertion of the LTR region of the human endogenous retrovirus 9 upstream of the TAp63 gene resulting in the addition of 37 amino acids at the N-terminus. A detailed biochemical analysis of this isoform has shown that this additional sequence results in a further stabilization of the dimeric, inactive state[16]. Why (human)

male germ cells require this extra safety mechanism remains an open question but it has been proposed that expression of GTAp63α is important to enable longer reproductive periods in hominids. The different roles that p63 isoforms play in female and male germ cell quality control also reflect the different strategies used to produce these cells. In the female germline stem cells are quiescent and a limited number of arrested germ cells constitutes the ovarian reserve. In contrast, in the male germline stem cells are actively dividing resulting in the mass production of sperm cells. Differences in quality surveillance are also manifested in the observation that disease causing point mutations disproportionally arise from the male germline, whereas chromosomal aberrations are mostly transmitted by female germ cells.

While TAp63β causes female infertility in HET Δ13p63, replacement of ΔNp63α with ΔNp63β in other tissues/organs does not cause morphological or functional defects. This is probably because ΔNp63α and ΔNp63β are tetramers and act synergistically to control target gene expression, acting both as repressors and activators. However, we must keep in mind that potential as-yet unidentified SAM-domain interactors or post-translational modifications, such as sumoylation/ubiquitylation occurring at the p63 C-terminal lysine residues[23,26,57,58], are abrogated in the ΔNp63β isoform, leaving open the possibility that under particular stress/pathological conditions, epithelial tissues lacking ΔNp63α could be defective.

Although TAp63 and ΔNp63 isoforms play roles in mouse embryo development and adult lifespan regulation, the relative contributions and roles of their C-termini have not been conclusively established in a defined in vivo genetic system. The generation and characterization of HET Δ13p63 mice clearly contributed to understand the molecular basis and the biological importance of TAp63 regulation for oocytes quality control. Further studies, based on exon 13 conditional deletions, will help in determing the role of p63 C-terminus in other p63-expressing tissues. Our results show that the TID inhibitory effect is indispensable in TAp63α-expressing primary oocytes to control cell death in vivo, supporting and expanding the current knowledge regarding human primary ovarian insufficiency.

## Methods

### Generation of Δ13p63 heterozygous mice and maintenance of the mouse colony.
P63 exon 13 floxed allele mice were generated by Ozgene (Ozgene Pty Ltd, Bentley, Australia), and the floxed/+ colony was amplified by crossing with wild-type C57Bl/6J mice. Heterozygous floxed/+ mice were crossed with Tg(CMV-cre) 1Cgn homozygous mice[59] to obtain the p63 exon 13-deleted allele (Δ13p63). The Δ13p63/CMV-Cre heterozygous mice were backcrossed with wild-type C57Bl/6J mice to lose the CMV-Cre allele. The pure Δ13p63 heterozygous mice obtained were crossed as described in Table S1. Transgenic p-70 GFP/c-Kit[60] female mice were crossed with Δ13p63 heterozygous male mice to obtain p-70 GFP/c-Kit Δ13p63 heterozygous ovaries. The mouse genotypes were assessed by PCR using the primers listed in Supplementary Table 1. The mice were bred in-house and housed in a temperature- and light-controlled mouse colony room (12-h light/dark cycle) and had free access to food and water. All experiments were approved by the Institutional Animal Care and Use Committee (IACUC) and were carried out according to the Italian and European rules (D.L.116/92; C.E. 609/86; European Directive 2010/63/EU). For mice experiments licence n° 817/2016PR (Italian Ministry of Health).

### Cell cultures, transfections, and luciferase assays.
HEK293 and H1299 cell lines were grown in Dulbecco's modified Eagle's medium (Thermo Fisher, Waltham, MA, USA) supplemented with 10% (vol/vol) foetal bovine serum (Thermo Fisher, Waltham, MA, USA) and 1% penicillin/streptomycin (Thermo Fisher, Waltham, MA, USA) at 37 °C in a humidified atmosphere of 5% (vol/vol) $CO_2$ in air.

A total of $1.2 \times 10^5$ HEK293 cells were seeded in 12-well dishes 24 h before transfection. A total of 100 ng of pGL3 vectors, 300 ng of transactivators expression vectors and 10 ng of Renilla luciferase pRL-CMV vector (Promega, Madison, WI, USA) were cotransfected using Effectene Reagent (Qiagen, Hilden, Germany) according to the manufacturer's protocol. pGl3 vectors containing p63-responsive elements in ENV and BPAG1 promoters[61].

*Puma* and *Noxa* p63-responsive elements described before[14] were amplified by PCR from human genomic DNA and cloned in pGL3basic vector (Promega, Madison, WI, USA). Relative luciferase activities were measured 24 h after transfection using a Dual Luciferase Reporter Assay System (Promega, Madison, WI, USA). Light emission was measured over 10 s using a Lumat LB9507 luminometer (EG&G Berthold, Wildbad, Germany). The efficiency of transfection was normalized to Renilla luciferase activity.

A total of $0.8 \times 10^5$ H1299 cells were seeded in a 12-well plate 24 h before transfection. 267 ng of vectors *Puma* 4xBS2WT-Luc[62] (kind gift from B. Vogelstein), pcDNA3, and pRL-CMV were cotransfected using Lipofectamine® 2000 according to manufacturer's manual and grown for 24 h. Relative luciferase activities were measured as described before. Measurement was performed at Tecan Spark multimode microplate reader (Tecan, Männedorf, Switzerland). A total of $4 \times 10^5$ H1299 cells were seeded 24 h before transfection. The same amount of HA tagged TAp63α and TAp63β expression vectors, or empty vector as control, were cotransfected using Effectene Reagent (Qiagen, Hilden, Germany) according to the manufacturer's protocol. 24 h after transfection, medium was removed and replaced with complete fresh medium. 10 μM MG132 or DMSO as control were added. After 1 h, cycloheximide treatment (50 μg/ml) was started and samples were collected at 0, 2, 4, 6, and 8 h.

Primary mouse keratinocytes were isolated from newborn mouse skin and collected in proliferation conditions.. Briefly newborn mouse pups were sacrificed, epidermises were separated from dermis by overnight incubation in trypsin, then they were minced with scissors, triturated by pipetting several times. Single cells were filtred through a 100-μm cell strainer (Corning, Glendale, AZ, USA), centrifuged at 700 rpm for 7' and finally plated on rat tail collagen (Corning, Glendale, AZ, USA) coated dishes in low calcium medium: EMEM 06-174G (LONZA Bioscience, Basel, Switzerland), 8% chelate FBS, antibiotic/antimycotic solution (Sigma, St. Louis, MO, USA) and 0,05 mM $CaCl_2$. Twenty-four hours later, aderent cells were washed three times in PBS and then collected to be processed as desired.

### RNA extraction and real-time PCR analysis.
Total RNA was isolated from mouse tissues by RNeasy Mini Kit (Qiagen, Hilden, Germany) and retrotranscribed by GoScript Reverse Transcription System (Promega, Madison, WI, USA) according to manufacturer's protocol. Real Time PCR was performed using GoTaq qPCR Master Mix (Promega, Madison, WI, USA). Primers used are listed in Supplementary Table 1. The expression of each gene was defined from the threshold cycle (Ct) and the relative expression levels were calculated by using the $2^{-\Delta\Delta Ct}$ method after normalization with reference to expression of β-actin as housekeeping gene. Endpoint semi-quantitative PCR were performed using GoTaq polymerase (Promega, Madison, WI, USA) according to manufacter protocol. Primers used are listed in Supplementary Table 1.

### Histology and immunofluorescence.
Freshly isolated mouse skin and thymus samples were embedded in OCT (Bio-Optica, Milano, Italy), cryosectioned (10 μm) and section fixed in 4% paraformaldehyde for 10' at room temperature. Testes and ovaries were fixed directly in 4% paraformaldehyde (12–16 h), then embedded in paraffin and H/E or PAS stained following standard procedures.

For immunofluorescence (IF) staining, cryosections (10 μm) were fixed in 4% paraformaldehyde for 10' at room temperature. Nonspecific antigens were blocked by incubation in 5% goat serum in PBS for 1 h in a humidified atmosphere at room temperature. Subsequently, the sections were incubated with primary antibodies for 2 h at room temperature. The sections were then washed three times with PBS and incubated for 1 h with the appropriate secondary antibodies conjugated with Alexa Fluor 488 or 568 (Thermo Fisher, Waltham, MA, USA) and DAPI. The slides were then mounted using Prolong Antifade (Thermo Fisher, Waltham, MA, USA) IF images were acquired with a Nikon A1 confocal laser microscope (Nikon, Tokyo, Japan).

For follicle counting, whole P1 ovaries were sectioned and stained with an anti-MSY2 primary antibody. All MSY2-positive oocytes were counted in every fifth section throughout the entire ovary, and the mean total number of oocytes/sections was multiplied by the total number of sections of each ovary.

The catalogue numbers and sources of primary antibodies were as follows: anti-K5 (PRB-160P BioLegend; San Diego, CA, USA; 1/1000), Troma-1 (anti-K8, Developmental Studies Hybridoma Bank, Iowa City, IA, USA; 1/2000), anti-K14 (PRB-155P BioLegend; San Diego, CA, USA; 1/1000), anti-K10 (PRB-159P BioLegend; San Diego, CA, USA; 1/1000), anti Ki67 (91295-D3B5 Cell Signaling Technologies, Danvers, MA, USA; 1/200), anti β4 integrin (611232, BD Biosciences, San Jose, CA, USA), anti-p63 (D9L7L, Cell Signaling Technologies, Danvers, MA, USA; 1/200), and Msy2 (A12 Santa Cruz Biotechnologies, Dallas, TX, USA; 1:2000).

### Ovary culture.
Ovaries were collected at P1 from p-70 GFP/c-Kit mice and cultured in tissue culture four-well plates in 0.3 ml of α-MEM supplemented with 10% FBS, L-glutamine, penicillin-G, streptomycin, pyruvic acid, N-acetyl-L-cysteine and ITS liquid medium supplement at 37 °C in 5% $CO_2$ for a maximum of 7 days; the medium was changed every 2 days. In the experiments in which the dynamics of oocytes degeneration was followed in the presence of 50 μM Z-VAD-FMK or DSMO, in order to favour the inhibitor diffusion, each ovary was sliced into four pieces and cultured for 4 days under the conditions reported above except that medium was changed every day. Ovaries and fragments were then analysed for 3D rendered confocal imaging. Z-stacks were acquired every 5 μm for a total of 300 μm, and 3D rendering was performed with NIS Elements software (Nikon, Tokyo, Japan) using the alpha-blending algorithm. To estimate the number of oocytes in the experiment with Z-VAD treatment, the sum intensity of GFP fluorescence per fragment was quantified for each timepoint (from P1 to P4) using NIS Elements software. Intensity values per fragment for each timepoint were normalised to the intensity values at P1. Pearson coefficient (R) was calculated to confirm the correlation between intensity of GFP fluorescence and oocytes number. Manually counted oocytes at P4 were used for correlation analysis.

### Western blotting.
Cell and tissues extracts were resolved on SDS polyacrylamide gels and blotted onto Hybond P, PVDF membrane (GE Healtcare Chicago, IL, USA). Membranes were blocked with PBST 5% non-fat dry milk, incubated with primary antibodies for 2 h at room temperature, washed and hybridized with peroxidase conjugated secondary antibodies for 1 h at room temperature (goat anti-rabbit or goat anti-mouse, Biorad Hercules, CA, USA). Detection was performed with the ECL chemiluminescence kit (Perkin Elmer, Waltham, MA, USA) The antibodies used were: anti-p63α (D2K8X, Cell Signaling Technologies, Danvers, MA, USA; 1/500 dilution), anti-p63 (D9L7L, Cell Signaling Technologies, Danvers, MA, USA; 1/250), anti-GAPDH (clone 6C5, Merck Millipore, Darmstadt, Germany), anti-HA.11 (Biolegend, San Diego, CA, USA; 1/500), and anti-βactin (AC-15, Sigma, St. Louis, MO, USA; 1/5000).

Uncropped blots for westerns and other blots can be found in the Source data file.

### In vitro and mammalian cell expression followed by blue native PAGE (BN-PAGE).
For BN-PAGE, the different p63 isoforms/mutants were expressed either in H1299 cells or using the TNT® T7 Coupled Reticulocyte Lysate System (Promega, Madison, WI, USA) following manufacturer's protocol. For expression in H1299 cells, cells were transfected, harvested and lysed in 1× BN-PAGE lysis buffer (50 mM Tris pH 8.0, 100 mM NaCl, 20 mM CHAPS, 0.5 mM TCEP, 2 mM $MgCl_2$, 1× cOmplete (Roche, Basel, Switzerland), and 1× PhosSTOP (Roche, Basel, Switzerland) with 1 μl Benzonase (Merck) added on ice for 1 h. The supernatant was supplemented with 3× Native PAGE sample buffer (60% glycerol v/v, 15 mM coomassie G250) and samples were analysed using the BN-PAGE Novex 3–12% Bis-Tris protein gel system (Life Technologies) according to the manufacturer's instructions. The cathode buffer was supplemented with 0.002% coomassie G250, and the separation was performed at 4 °C for 60 min at 150 V followed by 60 min at 250 V. The BN-PAGE was transferred onto a 0.45 μm PVDF membrane using a tank-blot method from Life Technologies (XCell II blot module) according to the manufacturer's instructions. Proteins were transferred at 30 V for 105 min at 4 °C with precooled 1× NuPAGE™ Transfer buffers. After the transfer, the membrane was incubated for 15 min in 8% acetic acid and afterwards destained using 100% methanol. For in vitro expression, 20 μl of reticulocyte lysate was mixed with 1 μg of DNA in a total volume of 25 μl. After 90' at 30 °C, 20 μl of the reaction was mixed with 100 μl of 2× native PAGE buffer in a total volume of 200 μl. Then, 10 μl

of this mixture was mixed with 5 µl of 3× native PAGE loading buffer (60% glycerol and 15 mM Coomassie G250). A total of 7.5 µl of each final sample was subsequently used for native PAGE and western blot analysis, as previously described[29]. The p63 isoforms /mutants were detected using an N-terminal Myc-Tag (anti-Myc antibody 4A6, Millipore, Burlington, MA, USA) or a p63-specific antibody (ab124762, Abcam, Cambridge, UK).

**In silico analysis of aggregation propensity**. TANGO algorithm (http://tango. crg.es/) was used to investigate the β-aggregation propensity of wild-type and developmental syndromic mutants of p63α. The following parameters were chosen for analyzation: pH 7.2, temperature of 4 °C and ionic strength 0.15 M.

**Statistics and reproducibility**. Data are shown as individual values and/or means ± Standard deviation (SD), if not otherwise stated. Experimental results significance was evaluated by the Student's $t$-test, differences with $p < 0.05$ were considered significant. Statistics in unpaired Student's $t$-test, Chi-square test and Pearson correlation analysis were performed by GraphPad Prism 7.0 software (GraphPad Software Inc., San Diego, CA, USA). All experiments were performed at least three times.

**Reporting summary**. Further information on research design is available in the Nature Research Reporting Summary linked to this article.

## Data availability

All data generated or analysed during this study are included in this published article and its Supplementary Information files. Source data are provided with this paper.

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

## Acknowledgements

This work has been mainly supported by AIRC Grants (IG-22206 to E.C.) and Ministry of Health and Fondazione Luigi Maria Monti IDI-RCCS (RC to E.C.). The work was also partially supported by AIRC (IG#20473, 2017-2022; Start-up ID23219, 2019-2024). V.D. acknowledges support by DFG (DO 545/20-1).

## Author contributions

A.M.L., V.R., S.O., A.S., C.O., M.T. and A.C. performed the experiments. A.M.L., V.R. and A.S. characterized mouse phenotype (skin, thymus, ovary), S.O., C.O. and M.T. performed transcription activity assays and oligomerization studies. E.C., A.M.L., F.G.K. and V.D. designed the research; A.M.L., V.R., S.O., A.S., C.O., A.C., F.G.K., M.A.P., I.A., H.V., M.D.F. and G.M. analysed the data; E.C. and V.D. wrote the paper; all the authors read the paper and made comments.

## Competing interests

The authors declare no competing interests.
