## [Peer Review File · Nature Communications]

Reviewers' Comments:

Reviewer #1:

Remarks to the Author:

It has been established that the proapoptotic transcription factor p63, particularly its N-terminal deletion isoform, plays a key role in mediating DNA damage-induced oocyte apoptosis. However, p63 is also expressed as several C-terminal isoforms, but the *in vivo* functions of these isoforms in postnatal ovaries have not been determined. Therefore, in this study, Lena AM and coauthors investigated the role of P63beta isoform in regulating oocyte survival during ovarian follicle development. To do this, they generated a mouse model in which the exon 13 was deleted in the p63 gene, so that p63alpha isoform was replaced by p63beta. The resultant heterozygous females were infertile due to prompt loss of oocytes within a week after birth, due to the ectopic expression of the constitutively active p63beta in female germ cells. The phenotype resembles the premature ovarian insufficiency (POI) in women. Further investigations indicated that some mutations found in human p63alpha affected the oligomeric stage of p63alpha, and therefore induce POI as p63beta does.

The data are generally convincing and support all of the major conclusions. In addition, this study has apparent clinical relevance to human infertility. However, while the importance of p63 activity in maintaining ovarian integrity has already been demonstrated, the current results provided detailed but not breakthrough knowledge to the field.

The experiments in the manuscript were well performed and appropriately interpreted. Therefore I only have some minor comments and indicated several editing errors for further improvement.

Specific comments:

The Abstract can be better organized. For example, giving brief background introduction in the beginning and then summarize the major results and conclusions of this study. It is kind of confusing to insert extra background narrations ("Several human syndromes are caused by ... have so far been mostly neglected.") in the middle of result description.

The Introduction is too lengthy and need to be considerably shortened. For example, some functions of p63 isoforms in tissues other than the reproductive system are unnecessary to be described because this is not a review paper. A great deal of information is provided in the Introduction, but the authors did not emphasize the major points.

Page 5: "the presence only of the TAp63 isoform in WT ovaries (Fig 3b)." It should be Fig. 2b.

Page 6: "we observed significant upregulation of apoptotic TAp63 target genes involved in primordial follicle cell death, namely, PUMA and NOXA." When gene names being mentioned, they should be in italic, and only the initial letter is in capital (Puma and Noxa).

Supplementary Fig. 4a: "WB analysis of TAp63 isoforms expression in the luciferase assay extracts described in Fig. 5c." But Fig. 5c does not show luciferase assay results. I think it should be "Fig. 4d". The legend for Supplementary Fig. 4b is also incorrect.

Figure 2b: can the authors give some explanation why p63 mRNA level decreased in the ovaries of HET animals at P7?

Reviewer #2:

Remarks to the Author:

This paper characterises the c terminus of p63 in oocytes. To do this, a novel genetic mouse model (HET Δ 13p63 mice) was studied in which the p63 α isoform was replaced with the p63 β isoform. Female heterozygous mice were infertile due to widespread oocyte apoptosis resulting from increased expression of Puma and Noxa mediated by the constitutively active TAp53 β isoform. Notably, the data presented in this paper suggest that the transactivation inhibitory domain (TID) present in the TAp63 α isoform is important for controlling cell death. The major conclusion is that the c-terminus of p63 has an essential role for the maintenance of oocyte number by suppressing apoptosis, and thus certain mutations that compromise the c-terminus (and the TID in particular) may impair fertility in women.

The paper is well written, scientifically sound and reports an important finding that will be of interest to those working in discovery science (gamete biology, apoptosis, cell biology) and clinical medicine (reproductive medicine and fertility). This paper makes an important and significant contribution to the field.

I have only a few very minor issues for the authors to address:

1. Line 198 "Interestingly, TAp63 is dispensable for ovarian development, as shown in TAp63 KO mice; nevertheless, it plays a crucial role in the quality control of the primary oocytes being activated and inducing apoptosis in those suffering irreparable DNA damage". I suggest the authors delete the word irreparable as a number of studies suggest that the oocytes can indeed repair DNA damage very efficiently.
2. Line 241 "Under these conditions, we observed a significant reduction in cell death, as indicated by the rescue of the green fluorescent oocytes in Z-VAD-treated HET Δ 13p63-GFP ovaries compared to control DMSO-treated HET Δ 13p63-GFP ovaries 243 (Fig. 3b-c)." I am a little surprised by this result. I might have expected a delay in death when caspases were inhibited, but not a reduction, and indeed I think this is what the data in figure 4 shows. Can the authors modify the wording in the text to reflect this?
3. Counting the green dots/ovary fragments as shown in Figure 4b/c is potentially problematic. I am not sure how it is possible to distinguish one dot from another if they are very close together (as oocytes tend to be in neonatal ovaries). It would be impossible to tell if one dot represented 1 or 2 or even three oocytes. Can the authors describe in the methods how these dots were actually counted and comment on how confident they are about these data?
4. Figure 1 E- how many males and females were in this trial. Please include this information in the legend.
5. Can a more detailed description be given in the legend for figure 5f, it is very brief and needs more information to enable the schematic to be easily interpreted.
6. Much of the discussion contains material better suited to the introduction e.g. the entire second paragraph is background detail. This section should be reworked to provide a discussion of the strengths and limitations of the results in the paper in the context of current knowledge.
7. As a general comment, very very small sample sizes were used in this study. In most cases $n=3$.

Reviewer #3:

Remarks to the Author:

In this manuscript, authors investigated a completely unstudied p63 isoform, p63beta, using an in vivo mouse model for its biological function and biochemical analyses for its molecular properties. Their compelling data demonstrated that heterozygous mice with both p63 alpha and beta isoforms have normally developed epithelia but severely affected primary ovary. With their solid biochemical analyses, authors suggest that the constitutively active TAp63beta tetramer may be the trigger of uncontrolled cell death that leads to primary ovary insufficiency. The manuscript is well written. The work is novel, and may be relevant for female infertility.

However, some of the observations on p63beta expression in ovary are puzzling. Better analyses and interpretations are essential for understating the function of p63 beta.

1) In the ovary of p63alpha/beta heterozygous mice, both isoforms are strongly expressed in P1 but lowly expressed in P7 at the mRNA level. At the protein level, TAp63beta is not detectable even in P1. Authors reasoned that this is due to the rapid degradation of TAp63beta. However, if TAp63beta is very fast degraded, how can it play a significant role in apoptosis. This is contradicting to what authors have describe 'constitutively active isoform expression'. What is the mechanism of rapid degradation of TAp63beta. Is sumoylation involved, as authors showed previously for the p63 C-terminal mutants? If authors can inhibit beta degradation, e.g. inhibit sumoylation, do they expect to see a greater effect of apoptosis? What is the protein expression level of TAp63beta in P7?

2) The most important question is the difference in time for the observed phenotypes and p63alpha

and beta expression. In the manuscript, phenotypic differences were observed at later stages of ovary development: size difference in P45 (Fig. 2a), cell death difference from P3 onwards (Fig. 3a). Relevant to the first point, TAp63alpha expression is already very low in P1 and beta is barely detectable. What are their gene expression at later stages? Is it possible that TAp63alpha/beta (if beta is expressed at all) initiate apoptosis very early on, even before P1, and the cell death authors observed controlled by downstream cascade, rather than by p63 isoforms themselves. Also relevant here, which stage was tested for Puma and Nova expression shown in Fig. 3d?

Minor points

1. In Fig. 1e, only the pups number was shown, but not the data on Mendelian ratio, as authors suggested in the text. It would be good to include the data.
2. In Fig. 3b, how good is the effect of Z-VAD in rescue, as compared to the WT situation?
3. In their biochemical analyses, some C-terminal mutants (e.g. InsA1572, DelAA1743) performed as well as TAp63beta in transactivation and oligomerization, even better than those that have been associated with female infertility (e.g. R555, W559). It is fair to suggest that female infertility should be tested in these patients and probably also in some AEC patients, and the p63 gene can be tested in female infertility patients.

Point-by-point reply to REVIEWERS

REF: re-submission Paper NCOMMS-20-05029

"The p63 C-terminus is essential for oocytes integrity"

Lena AM et al. (Corresponding: G Melino & E Candi)

We thank the Referees for their positive comments and for their constructive criticisms that allowed us to produce an improved revised manuscript.

Questions raised by the REFEREES are in italic blue colour

REVIEWER 1

REVIEWER 1: *It has been established that the proapoptotic transcription factor p63, particularly its N-terminal deletion isoform, plays a key role in mediating DNA damage-induced oocyte apoptosis. However, p63 is also expressed as several C-terminal isoforms, but the in vivo functions of these isoforms in postnatal ovaries have not been determined. Therefore, in this study, Lena AM and coauthors investigated the role of P63beta isoform in regulating oocyte survival during ovarian follicle development. To do this, they generated a mouse model in which the exon 13 was deleted in the p63 gene, so that p63alpha isoform was replaced by p63beta. The resultant heterozygous females were infertile due to prompt loss of oocytes within a week after birth, due to the ectopic expression of the constitutively active p63beta in female germ cells. The phenotype resembles the premature ovarian insufficiency (POI) in women. Further investigations indicated that some mutations found in human p63alpha affected the oligomeric stage of p63alpha, and therefore induce POI as p63beta does.*

The data are generally convincing and support all of the major conclusions. In addition, this study has apparent clinical relevance to human infertility. However, while the importance of p63 activity in maintaining ovarian integrity has already been demonstrated, the current results provided detailed but not breakthrough knowledge to the field. The experiments in the manuscript were well performed and appropriately interpreted. Therefore I only have some minor comments and indicated several editing errors for further improvement.

Reply: We thank the referee for the positive and constructive comment. Indeed, contrary to the previous genetic ablation, the replacement p63alpha with p63beta is much more subtle and delicate, providing a more convincing evidence. We hope this reviewer would appreciate this new version.

REVIEWER 1: *The Abstract can be better organized. For example, giving brief background introduction in the beginning and then summarize the major results and conclusions of this study. It is kind of confusing to insert extra background narrations ("Several human syndromes are caused byhave so far been mostly neglected.") in the middle of result description.*

Reply: Thank you for pointing this. We have now modified the abstract as suggested by the referee. We omitted the sentence ("Several human syndromes are caused byhave so far been mostly neglected") and slightly modified the abstract for a more linear presentation of the article content.

REVIEWER 1: *The Introduction is too lengthy and need to be considerably shortened. For example, some functions of p63 isoforms in tissues other than the reproductive system are unnecessary to be described because this is not a review paper. A great deal of information is provided in the Introduction, but the authors did not emphasize the major points.*

Reply: As suggested by the referee, we have reduced the length of the introduction, giving emphasis to the major points, that is the current knowledge on p63 isoforms and the role of p63 in oocytes biology.

REVIEWER 1: *Page 5: “the presence only of the TAp63 isoform in WT ovaries (Fig 3b).” It should be Fig. 2b. Page 6: “we observed significant upregulation of apoptotic TAp63 target genes involved in primordial follicle cell death, namely, PUMA and NOXA.” When gene names being mentioned, they should be in italic, and only the initial letter is in capital (Puma and Noxa). Supplementary Fig. 4a:” WB analysis of TAp63 isoforms expression in the luciferase assay extracts described in Fig. 5c.” But Fig. 5c does not show luciferase assay results. I think it should be “Fig. 4d”. The legend for Supplementary Fig. 4b is also incorrect.*

Reply: We thank the referee for raising these points. We have amended all these errors in the revised version. We have now checked more carefully all reference to figures.

REVIEWER 1: *Figure 2b: can the authors give some explanation why p63 mRNA level decreased in the ovaries of HET animals at P7?*

Reply: We thank the referee for raising this point, giving us the opportunity to clarify p63 expression levels during ovaries development. We expanded the time-course, including day 3 and day 10 post-natal (Figure 2b) and analyzed p63 expression at mRNA (Figure 2b) and protein (immunofluorescence and confocal analysis, Figure 2g) level. In line with previous reports (Suh *et al*, 2006), in WT mice p63 expression is detectable at embryonic stage E17.5, increases at day 1 and its expression remains high in primordial and primary follicles (P1 to P10.) In HET mice, at later time points, the p63 mRNA level decreases because, as soon as the p63beta isoform is expressed, oocytes die by apoptosis (see Figure 2d-g, Figure 3). Since within the ovary p63 is exclusively expressed in the oocytes, loss of oocytes due to cell death causes loss of p63 mRNA. We clarify this point in the main text as follows:

“Through semiquantitative RT-qPCR analysis of TAp63 expression in ovaries from WT and HET $\Delta 13p63$ females from day 17.5 of embryonic development (E17.5) to postnatal days 1 (P1), 3 (P3), 7 (P7) and 10 (P10), we confirmed that in WT mice TAp63 α isoform is detectable at E17.5 and that its expression remained high in primordial and primary follicles (P1 to P10; Fig. 2b). On the other hand, in HET $\Delta 13p63$ ovaries, while there was the expected increase from E17.5 to P1 (Fig. 2b), at later time points, both p63 α and p63 β mRNA levels decreased along with a reduction in oocytes number (Fig. 2d-g), which appeared associated to increased apoptosis (Fig. 3). Since within the ovaries, p63 is exclusively expressed in the oocytes, depauperation of these population resulted in reduction of p63 mRNA. In HET $\Delta 13p63$ ovaries, p63 γ isoform and p53 mRNAs followed similar trend (Supplementary Fig. 1e). Western blot analysis showed that only the TAp63 α isoform was detectable at very low levels in P1 HET $\Delta 13p63$ ovary extracts, while the levels of the TAp63 β variant were not appreciable (Fig. 2c). Mechanistically, however we also

proved that active TAp63 β variant undergoes a high proteosomal degradation rate (Supplementary Fig. 2a-c). Thus, both oocytes depauperation and poor protein stability underlined low expression level of TAp63 β in HET Δ 13p63 ovary extracts.”

REVIEWER 2

REVIEWER 2: *This paper characterises the c terminus of p63 in oocytes. To do this, a novel genetic mouse model (HET Δ 13p63 mice) was studied in which the p63 α isoform was replaced with the p63 β isoform. Female heterozygous mice were infertile due to widespread oocyte apoptosis resulting from increased expression of Puma and Noxa mediated by the constitutively active TAp53 β isoform. Notably, the data presented in this paper suggest that the transactivation inhibitory domain (TID) present in the TAp63 α isoform is important for to controlling cell death. The major conclusion is that the c-terminus of p63 has an essential role for the maintenance of oocyte number by suppressing apoptosis, and thus certain mutations that compromise the c-terminus (and the TID in particular) may impair fertility in women. The paper is well written, scientifically sound and reports an important finding that will be of interest to the those working in discovery science (gamete biology, apoptosis, cell biology) and clinical medicine (reproductive medicine and fertility). This paper makes an important and significant contribution to the field. I have only a few very minor issues for the authors to address.*

Reply: We thank the referee for the positive comments and for highlighting the potential interest of our manuscript in the field of reproductive biology. We really appreciate.

REVIEWER 2: *1. Line 198 “Interestingly, TAp63 is dispensable for ovarian development, as shown in TAp63 KO mice; nevertheless, it plays a crucial role in the quality control of the primary oocytes being activated and inducing apoptosis in those suffering irreparable DNA damage”. I suggest the authors delete the word irreparable as a number so studies suggest that the oocytes can indeed repair DNA damage very efficiently.*

Reply: We thank the referee for raising this point. We have omitted the word “irreparable” in the revised version.

REVIEWER 2: *2. Line 241 “Under these conditions, we observed a significant reduction in cell death, as indicated by the rescue of the green fluorescent oocytes in Z-VAD-treated HET Δ 13p63-GFP ovaries compared to control DMSO-treated HET Δ 13p63-GFP ovaries 243 (Fig. 3b-c).” I am a little surprised by this result. I might have expected a delay in death when caspases were inhibited, but not a reduction, and indeed I think this is what the data in figure 4 shows. Can the authors modify the wording in the text to reflect this?*

Reply: We thank the reviewer for this suggestion. Indeed, we intended to highlight the delay of the cell death induced by Z-VAD-mediated inhibition of caspases. We amended the main text as follows:

“Under these conditions, we observed a significant delay in cell death, as indicated by the rescue of the green fluorescent oocytes in Z-VAD-treated HET Δ 13p63-GFP ovaries compared to control DMSO-treated HET Δ 13p63-GFP ovaries (Fig. 3b-c).”

REVIEWER 2: *3. Counting the green dots/ovary fragments as shown in Figure 4b/c is potentially problematic. I am not sure how it is possible to distinguish one dot from another if they are very close*

together (as oocytes tend to be in neonatal ovaries). It would be impossible to tell if one dot represented 1 or 2 or even three oocytes. Can the authors describe in the methods how were these dots actually counted and comment on how confident they are about these data?

Reply: We appreciate reviewer's concerns about the quantification of number of oocytes in Figure 3b/c experiment. Since it is very difficult to distinguish the single oocytes (green dots) during the first days after birth (P1-P2), as kindly highlighted by the reviewer, we were unable to quantify the exact number of oocytes. Therefore, we employed an indirect way of estimation by quantifying the sum intensity of GFP per ovary fragment using NIS Elements software (Nikon). To confirm that this way of quantification gives adequate results, we performed correlation analysis between the intensity of GFP per fragment and exact number of oocytes. To this end, we manually counted the number of oocytes in DMSO or Z-VAD treated samples from P4 ($n=14$), where the oocytes are clearly separated, and afterwards quantified GFP intensity with Nikon software. We modified the Figure 3c by adding a scatterplot showing correlation between GFP intensity and number of oocytes from P4 fragments. As can be observed, there is a strong positive correlation between intensity of GFP and number of oocytes (Pearson correlation $R=+0.90$, $P=1.3 \times 10^{-5}$):

Thus, the quantification of GFP intensity in other samples (P1 to P3) can indirectly provide the information of estimated number of oocytes. Since there is high variability in initial number of oocytes per fragment, we normalised all intensity values to the intensity value of fragment at P1 (equal to one). We agree that description of this quantification is quite poor and might be confusing. Therefore, we changed the Y-axis title of the graph in Figure 3c from “Green dots count/ovary fragments (fold over P1)” to “Intensity of GFP per ovary fragment (fold over P1)”. Moreover, we expanded the “Ovary culture” section in Material and Methods as follows:

“...was performed with NIS Elements software (Nikon, Tokyo, Japan) using the alpha-blending algorithm. To estimate the number of oocytes in the experiment with Z-VAD treatment, the sum intensity of GFP per fragment was quantified for each time point (from P1 to P4) using NIS Elements software. Intensity values per fragment for each time point were normalised to the intensity values at P1. Pearson coefficient was calculated to confirm the correlation between intensity of GFP and number of oocytes. Manually counted oocytes at P4 were used for correlation analysis.”

Accordingly, we amended the main text:

“To test whether the primary oocytes died by overt apoptosis, we added the pan-caspase inhibitor Z-VAD to the HET $\Delta 13p63$ -GFP ovary culture medium. To note, distinguishing of single oocytes immediately after birth (P0-P3) is quite challenging, making virtually impossible to count them. Therefore, we employed the quantification of intensity of GFP fluorescence to estimate the number of oocytes in different conditions. Indeed, intensity of GFP is proportional to the number of oocytes, as assessed by manual counting of oocytes at P4 where they are clearly separated. Interestingly, after Z-VAD treatment we observed a significant delay in cell death, as indicated by the rescue of the green fluorescent oocytes in Z-VAD-treated HET $\Delta 13p63$ -GFP ovaries compared to control DMSO-treated HET $\Delta 13p63$ -GFP ovaries (Fig. 3b-c).”

REVIEWER 2: 4. *Figure 1 E- how many males and females were in this trial. Please include this information in the legend.*

Reply: We thank the reviewer for this suggestion. We have indicated now in Figure 1 legend males and females mice numbers used in the experiment shown in Figure 1e.

REVIEWER 2: 5. *Can a more detailed description be given in the legend for figure 5f, it is very brief and needs more information to enable the schematic to be easily interpreted.*

Reply: We thank the referee for this suggestion. We modified Figure 5f description as follows:

“5f. Schematic model of TAp63 role in ovary physiology, in HET $\Delta 13p63$ mouse model and in presence of POI mutations. Under normal physiological conditions, TAp63 α is present in inactive dimeric form, which can be activated only upon certain circumstances such as DNA damage. Activated TAp63 α enables DNA quality check during to ensure the genomic stability of germline ovary development and follicle maturation. However, TAp63 β as well as POI mutants of p63 are active tetramers which constitutively induce cell death, leading therefore to premature ovary insufficiency.”

REVIEWER 2: 6. *Much of the discussion contains material better suited to the introduction e.g. the entire second paragraph is background detail. This section should be reworked to provide a discussion of the strengths and limitation of the results in the paper in the context of current knowledge.*

Reply: We thank the referee for this suggestion. As requested, we have eliminated the second paragraph in the revised version and modified the discussion to clarify the relevance of the results in the context of the current knowledge. We hope that the new Introduction and Discussion meet the approval of the reviewer.

REVIEWER 2: 7. *As a general comment, very very small sample sizes were used in this study. In most cases n=3.*

Reply: We thank the referee for giving us the opportunity to clarify this point. Due to the limited number of HET females obtained, we have used the minimal number of breeding mice that gave us a statistically significant result. Yet, to satisfy the referee requests, we decided to increase the number of

mice used in selected experiments. Specifically, we increased the number of mice analyzed for the experiment shown in Figure 2e from n=3 to n=6.

For the other *in vivo* experiments, we make sure that the sample size used was clearly stated in the figure legends. For instance, in Figure 1 and Figure 3 legends the number of mice analyzed is now clearly indicated; respectively, n=4 to n=10 and n=7 to n=14. We hope that, having increased the sample size of the experiment shown in Figure 2e and clearly specified the number of the mice used in the experiments shown in Figure 1 and Figure 3, will satisfy the referee's request.

REVIEWER 3

***REVIEWER 3:** In this manuscript, authors investigated a completely unstudied p63 isoform, p63beta, using an in vivo mouse model for its biological function and biochemical analyses for its molecular properties. Their compelling data demonstrated that heterozygous mice with both p63 alfa and beta isoforms have normally developed epithelia but severely affected primary ovary. With their solid biochemical analyses, authors suggest that the constitutively active TAp63beta tetramer may be the trigger of uncontrolled cell death that leads to primary ovary insufficiency. The manuscript is well written. The work is novel, and may be relevant for female infertility. However, some of the observations on p63beta expression in ovary are puzzling.*

Reply: We thank the referee for the positive comments and for highlighting the potential interest of our manuscript in the field of female infertility.

***REVIEWER 3:** 1) In the ovary of p63alfa/beta heterozygous mice, both isoforms are strongly expressed in P1 but lowly expressed in P7 at the mRNA level. At the protein level, TAp63beta is not detectable even in P1. Authors reasoned that this is due to the rapid degradation of TAp63beta. However, if TAp63beta is very fast degraded, how can it play a significant role in apoptosis. This is contradicting to what authors have describe 'constitutively active isoform expression'. What is the mechanism of rapid degradation of TAp63beta. Is sumoylation involved, as authors showed previously for the p63 C-terminal mutants? If authors can inhibit beta degradation, e.g. inhibit sumoylation, do they expect to see a greater effect of apoptosis? What is the protein expression level of TAp63beta in P7?*

Reply: We thank the referee for the opportunity to clarify in the article this important point. As mentioned above (referee1, point 5) protein and mRNA steady state level of p63 rapidly declines when oocytes start dying by apoptosis. We can hypothesize two reasons: either a simple loss/death of the p63-expressing cells, or a specific mechanism of degradation despite/after the transcriptional activation. Since p63 is exclusively expressed in the oocytes within the ovaries, loss of oocytes leads to the decline of p63 mRNA and protein, in absolute levels. In other words, the levels decrease because there are no more oocytes. In addition, a second (possibly minor) mechanism might be involved: activated p63 gets degraded by E3 ligases which constitutes a safety mechanism (Ying et al., 2005). We previously investigated this mechanism and set up model calculations. Results are published online in Browne et al., 2019; Gebel et al., 2016, Gebel et al 2019. This explains why level of TAp63 β are always lower than TAp63 α in WB on HET ovaries. The apparent contradiction that p63 β kills despite being degraded opens other questions: how long p63 β needs to be transcriptionally active to kill? How many molecules should be transcribed? Are few minutes with few molecules sufficient to

kill? Figure 4 and figure S4 show clearly the active tetramer configuration of TAp63 β (as well of clinically relevant mutants) whilst TAp63 α is in a dimeric form.

Furthermore, it has been reported previously that transcriptionally inactive (TAp63 α) or less active (Δ Np63) isoforms accumulate in cells while transcriptionally active isoforms are degraded fast (Ying *et al.*, 2005; Serber *et al.*, 2002). Since a positive feedback loop exists, only low levels of p63 protein are necessary to drive p63-dependent transcription.

To satisfy the referee request, however, we performed experiments using the protease inhibitor MG132 following cycloheximide treatments (Supplementary Figure 2a-c) to evaluate steady-state protein stability and confirm the involvement of proteasome system in TAp63 β degradation. The data obtained, taking into account the limits of the experimental model, confirmed the previous published finding (Ying *et al.*, 2005; Serber *et al.*, 2002; Browne *et al.*, 2011), indicating that the active TAp63 β variant is less stable than TAp63 α and that its degradation is proteasomal-dependent; still it can kill rapidly.

Supplementary Fig 2. TAp63 β is degraded faster than TAp63 α . (a) Schematic drawing of the experiment. H1299 cells were transfected with HA-tagged TAp63 isoforms and treated with MG132 and cycloheximide at the indicated time points. (b) Western blot was performed at different time points using anti-HA antibody. Beta-actin was used as loading control. Western blot shown is one of four experiments (n=4). (c) Average densitometric analysis of the western blots. Beta actin was used to normalize the values. Data are shown as relative level (%) over initial time point. Data are shown as mean \pm SEM. *p<0.05.

Finally, which E3 ligases are involved in the degradation of p63 after activation and the role of sumoylation, are active and highly interesting research projects and are far beyond the scope of the current manuscript. We modified the text as following:

In the paragraph: **Δ 13p63 heterozygous females show primary ovary insufficiency:**

“Through semiquantitative RT-qPCR analysis of TAp63 expression in ovaries from WT and HET Δ 13p63 females at day 17.5 of embryonic development (E17.5) and at postnatal days 1 (P1), 3 (P3), 7 (P7) and 10 (P10), we confirmed that in WT mice TAp63 α isoform is detected at E17.5 and that its expression remained high in primordial and primary follicles (P1 to P10; Fig. 2b). On the other hand, in HET Δ 13p63 ovaries, while there was the expected increase from E17.5 to P1 (Fig. 2b), at later time points, both p63 α and p63 β mRNA levels decreased along with a reduction in oocytes number (Fig. 2d-g), which appeared associated to increased apoptosis (Fig.3). Since within the ovaries, p63 is exclusively expressed in the oocytes, depauperation of these population resulted in reduction of p63 mRNA. In HET Δ 13p63 ovaries, p63 γ isoform and p53 mRNAs followed similar trend (Supplementary Fig. 1e). Western blot analysis showed that only the TAp63 α isoform was detectable at very low levels in P1 HET Δ 13p63 ovary extracts, while the levels of the TAp63 β variant were not appreciable (Fig. 2c). Mechanistically, however we also

proved that active TAp63 β variant undergoes a high proteosomal degradation rate (Supplementary Fig. 2a-c). Thus, both oocytes depauperation and poor protein stability underlyed low expression level of TAp63 β in HET Δ 13p63 ovary extracts.”

In the paragraph: **TAp63 β forms constitutively active tetramers**

“...TAp63 α and TAp63 β showed high transactivation in an in vitro transcription luciferase assay using the PUMA-responsive element in agreement with earlier reports (Fig. 4b, Supplementary Fig. 4a) ^{4,23}. To note, that the variants with high transcriptional activity are the one detected less efficiently by western blot (Fig. 4b).”

REVIEWER 3: 2) *The most important question is the difference in time for the observed phenotypes and p63 α and beta expression. In the manuscript, phenotypic differences were observed at later stages of ovary development: size difference in P45 (Fig. 2a), cell death difference from P3 onwards (Fig. 3a). Relevant to the first point, TAp63 α expression is already very low in P1 and beta is barely detectable. What are their gene expression at later stages? Is it possible that TAp63 α /beta (if beta is expressed at all) initiate apoptosis very early on, even before P1, and the cell death authors observed controlled by downstream cascade, rather than by p63 isoforms themselves. Also relevant here, which stage was tested for Puma and Noxa expression shown in Fig. 3d?*

Reply: We thank the referee for the opportunity to further clarify this important point. This comment has allowed us to rationalize the data and better present them in the current manuscript. We realized that the previous version might convey the concept that phenotype and expression timing of TAp63 isoforms differs. With the newly added data, however, we believe this apparent discrepancy has now been clarified. Expression of TAp63 α in oocytes physiologically starts at ~E18 (Figure 2b,c and *Suh et al., 2006*). In HET oocytes, apoptosis is initiated immediately after TAp63 β is expressed since this isoform is constitutively active. Consequentially, in P1 mice the number of oocytes starts already to be reduced in the HET when compared to WT mice (Figure 2d). Accordingly, Puma and Noxa mRNA levels are significantly upregulated (HET ovaries at stage P1, Fig 3d), indicating that the apoptotic process is indeed initiated when TAp63 β starts to be expressed, as expected. The gross morphology defects (reduced ovary size, etc) are macroscopically evident later on in the process, when the pool of oocytes has been completely depleted (stage P45).

The reduced number of oocytes will consequentially also lead to a reduction of the p63 level detected by western blot (p63 is reduced in HET vs WT P1 ovaries). Therefore, as correctly noted by the reviewer, at P1 the level of TAp63 β is already reduced. At P3 basically no oocytes are left due to apoptosis and hence, no p63 signal can be detected as p63 is only expressed in the oocytes within the ovaries. This was further demonstrated in the revised Figure 2 in which we have expanded the time points (stages P1, P5, P5 and P10) and analyzed (i) morphology (H/E staining, Figure 2b) and (ii) oocytes (immunofluorescence for MSY2 and p63) (Figure 2g). Ovaries without oocytes shrink, which is visible at P45.

Reduction in TAp63 β is however also associated to a reduced stability (see previous point and references, Supplementary Figure 2a-c). This also explains why in HET ovaries the TAp63 β level is lower than the already low TAp63 α level, which is probably reduced only for the decreased number of oocytes). The different stability properties of TAp63 β and TAp63 α are clearly shown in all the in vitro experiments performed. Indeed, western blots performed after 1:1 transfection ratio of TAp63 α :TAp63 β

variants, always showed very low level of TAp63 β , at steady-state conditions, see Figure 4b, Supplementary Figure 2b, Supplementary Figure 4b.

In conclusion, with these new data we believe that the timing for activation of apoptosis (Fig. 3d), reduction of number of oocytes (Fig. 2d) and appearance and decline of TAp63 β (Fig. 2b,c) are proven to be entirely consistent.

REVIEWER 3: 1. *In Fig. 1e, only the pups number was shown, but not the data on Mendelian ratio, as authors suggested in the text. It would be good to include the data.*

Reply: As correctly requested by the referee, we have indicated in the revised Figure 1b a Table showing the number of mice analyzed to establish the Mendelian ratio of newborn mice.

REVIEWER 3: 2. *In Fig. 3b, how good is the effect of Z-VAD in rescue, as compared to the WT situation?*

Reply: We thank the reviewer for this interesting question. As can be seen from images in Figure 3b and the quantification in Figure 3c, the treatment with Z-VAD was only able to delay the cell death (after P4 no significant rescue effects of Z-VAD were observed) rather than completely inhibit it. In part, it could be explained by the asynchrony of the cell death induction *in vivo* and *ex vivo*. Here, we use the *ex vivo* cultures of HET Δ 13p63 ovaries in which p63 β presumably triggers the apoptosis between E18.5 and P1 in asynchronous fashion which can explain why Z-VAD protects only some oocytes from cell death when added at P0.

REVIEWER 3: 3. *In their biochemical analyses, some C-terminal mutants (e.g. InsA1572, DelAA1743) performed as well as TAp63beta in transactivation and oligomerization, even better than those that have been associated with female infertility (e.g. R555, W559). It is fair to suggest that female infertility should be tested in these patients and probably also in some AEC patients, and the p63 gene can be tested in female infertility patients.*

Reply: We thank the reviewer for this interesting question. We are in contact with gynecologists and fertility clinics. But these are data requiring tissue and data from human patients. Which is subject to lengthy application procedures. Yet, the prediction of the analysis of these mutants for fertility of human patients with p63-based syndromes is quite clear and will be further analyzed in the future.

References

- 1) Suh E-K et al. p63 protects the female germ line during meiotic arrest. *Nature*. 2006. 444:624-628.
- 2) Ying H et al., DNA-binding and transactivation activities are essential for TAp63 protein degradation. *Mol Cell Biol*. 2005. 25(14):6154-64.
- 3) Browne, G. et al. Differential altered stability and transcriptional activity of DNp63 mutants in distinct ectodermal dysplasias. *J. Cell Sci*. 124, 2200–2207 (2011).
- 4) Gebel J et al. Mechanism of TAp73 inhibition by Δ Np63 and structural basis of p63/p73 hetero-tetramerization. *Cell Death Differ*. 2016. 23(12)1930-1940.
- 5) Gebel J et al. p63 sets the threshold for induction of apoptosis using a kinetically encoded ‘doorbell-like’ mechanism. 2019. bioRxiv 681007; doi: <https://doi.org/10.1101/681007>.
- 6) Serber, Z. et al. A C-Terminal Inhibitory Domain Controls the Activity of p63 by an Intramolecular Mechanism. *Mol. Cell. Biol*. 22, 8601–8611 (2002).

Reviewers' Comments:

Reviewer #1:

Remarks to the Author:

None

Reviewer #2:

Remarks to the Author:

I have no further comments. All of my queries have been appropriately addressed by the authors.

Karla Hutt

Reviewer #3:

Remarks to the Author:

Authors have comprehensively addressed my questions.